# CONTINUOUS ENSEMBLE WEATHER FORECASTING WITH DIFFUSION MODELS

**Martin Andrae**[1]  **Tomas Landelius**[2]  **Joel Oskarsson**[1]  **Fredrik Lindsten**[1]
[1]Linköping University, Sweden  [2]Swedish Meteorological and Hydrological Institute
{martin.andrae, joel.oskarsson, fredrik.lindsten}@liu.se
tomas.landelius@smhi.se

## ABSTRACT

Weather forecasting has seen a shift in methods from numerical simulations to data-driven systems. While initial research in the area focused on deterministic forecasting, recent works have used diffusion models to produce skillful ensemble forecasts. These models are trained on a single forecasting step and rolled out autoregressively. However, they are computationally expensive and accumulate errors for high temporal resolution due to the many rollout steps. We address these limitations with Continuous Ensemble Forecasting, a novel and flexible method for sampling ensemble forecasts in diffusion models. The method can generate temporally consistent ensemble trajectories completely in parallel, with no autoregressive steps. Continuous Ensemble Forecasting can also be combined with autoregressive rollouts to yield forecasts at an arbitrary fine temporal resolution without sacrificing accuracy. We demonstrate that the method achieves competitive results for global weather forecasting with good probabilistic properties.

## 1 INTRODUCTION

Forecasting of physical systems over both space and time is a crucial problem with plenty of real-world applications, including in earth sciences, transportation, and energy systems. A prime example of this is weather forecasting, which billions of people depend on daily to plan their activities. Weather forecasting is also crucial for making informed decisions in areas such as agriculture, renewable energy production, and safeguarding communities against extreme weather events. Current Numerical weather prediction (NWP) systems predict the weather using complex physical models and large supercomputers (Bauer et al., 2015). Recently Machine learning weather prediction (MLWP) models have emerged, rivaling the performance of existing NWP systems. These models are not physics-based but data-driven and have been made possible thanks to advancements in deep learning. By analyzing patterns from vast amounts of meteorological data (Hersbach et al., 2020), MLWP models now predict the weather with the same accuracy as global operational NWP models in a fraction of the time (Kurth et al., 2023; Lam et al., 2023; Bi et al., 2023).

Following the success of deterministic MLWP models, the focus of the field has increasingly shifted towards probabilistic modeling. The probabilistic models generate samples of possible future weather trajectories. By drawing many such samples, referred to as ensemble members, it is possible to generate a set of possible forecasts, referred to as an ensemble forecast, for quantifying forecast uncertainty and detecting extreme events (Leutbecher & Palmer, 2008). Sampling forecasts from deep generative models also address the blurriness often observed in predictions from deterministic MLWP, yielding forecasts that better preserve the variability of the modeled entities. A popular class of deep generative models used for probabilistic MLWP are diffusion models (Price et al., 2025; Lang et al., 2024; Larsson et al., 2025). While these models generate accurate and realistic looking forecasts, they are computationally expensive due to requiring multiple sequential forward passes through the neural network to generate a sample. Moreover, they are often applied iteratively to roll out longer forecasts (Price et al., 2025), which exacerbates the computational issue. Naively switching out this iterative rollout to directly forecasting each future timestep does not result in trajectories that are consistent over time. The auto-regressive rollout additionally puts some limitations on the temporal resolution of the forecast. Taking too small timesteps results in large accumulation of error over time (Bi et al., 2023), which forces existing models to resort to a temporal resolution

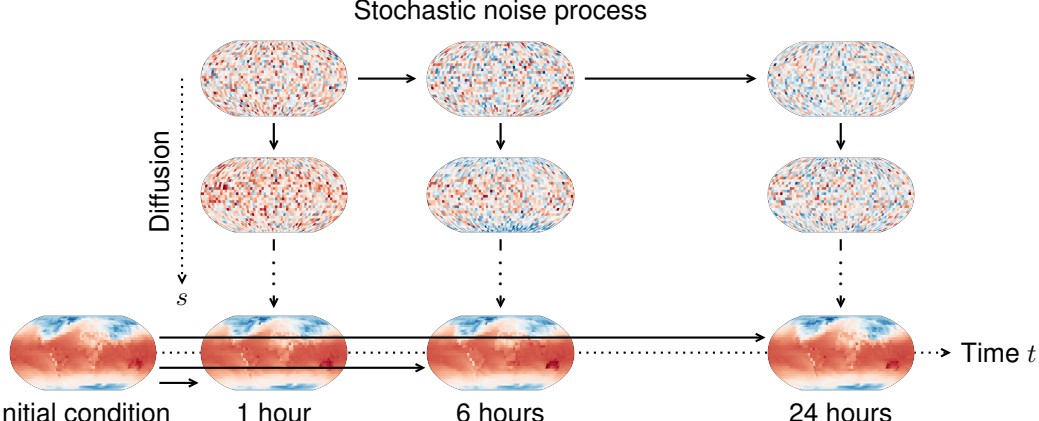

Figure 1: Our proposed framework, Continuous Ensemble Forecasting, generates ensemble weather forecasts using a conditional diffusion model. The model takes lead time as input and forecasts the future weather state in a single step, e.g. the forecast at 24 hours is generated directly from the initial condition, without seeing the intermediate predictions. To ensure temporal consistency, we correlate the driving noises for the different lead times. This can be done by fixing the noise, or by defining a stochastic noise process. Repeating the procedure for different starting noise gives an ensemble of forecasts. Using this framework we can generate 10 day forecasts with 1 hour resolution without sacrificing performance.

of 12h Price et al. (2025). However, in many situations it is crucial to obtain probabilistic forecasts at a much higher temporal resolution. This is true not least when the forecasts are used as decision support in extreme weather situations and for capturing rapidly changing weather events.

We propose a continuous forecasting diffusion model that takes lead time as input and forecasts the future weather state in a single step, while maintaining a temporally consistent trajectory for each ensemble member. [1] This enables both autoregressive and direct forecasting within a single model, improves the accuracy compared to purely autoregressive models at high temporal resolutions, and enables forecasting at arbitrary (non-equidistant) lead times throughout the forecast trajectory. To generate ensemble forecasts, the model solves the lead-time-dependent probability flow ODE starting in different pure noise samples. To ensure temporal consistency, we correlate the driving noises for the different lead times, e.g., by using a single noise sample for all timesteps, as illustrated in fig. 1, which enables generating a continuous trajectory for each member. This enables parallel sampling of individual ensemble members, bypassing the need for multi-step loss functions, and thus accelerating ensemble forecasting.

**Contributions.** We propose a novel method for ensemble weather forecasting built on diffusion that: 1) can generate ensemble member trajectories without iteration, 2) can forecast arbitrary lead-times, 3) can be used together with iteration to improve performance on long rollouts, 4) achieves competitive performance in global weather forecasting.

## 2 RELATED WORK

**Ensembles from perturbations.** In NWP, ensemble forecasts are created by perturbing initial conditions or model parameterizations. The same idea has been applied in MLWP using initial state perturbations (Bi et al., 2023; Kurth et al., 2023; Chen et al., 2023b; Li et al., 2024), model parameter perturbations (Hu et al., 2023; Weyn et al., 2021) and Monte Carlo dropout (Scher & Messori, 2021; Hu et al., 2023). In all these cases the underlying model is still deterministic, trained with a mean squared error loss function to predict the average future weather. These ensembles can thus be viewed as a mixture of means, inheriting the spatial oversmoothing characteristic of deterministic forecasts.

---

[1]The code is available at https://github.com/martinandrae/Continuous-Ensemble-Forecasting

**Generative models.** Another way of getting ensembles is by modeling the distribution directly through generative models. Price et al. (2025) train a graph-based diffusion model to produce 15-day global forecasts, at 12-hour steps and 0.25°resolution. Their GenCast model generates realistic forecasts but is slow to sample from, requiring sequential forward passes through the network both for sampling each timestep and to roll out the forecast over time. Latent-variable-based MLWP models (Hu et al., 2023; Oskarsson et al., 2024; Zhong et al., 2024) can perform inference in a single forward pass, but have been shown to suffer from more blurriness compared to diffusion models and require delicate hyperparameter tuning (Oskarsson et al., 2024). Apart from forecasting, diffusion models have also been successfully applied to other weather-related tasks, including downscaling (Chen et al., 2023a; Mardani et al., 2024; Pathak et al., 2024), enlarging physics-based ensembles (Li et al., 2024) and generating realistic weather from climate scenarios (Bassetti et al., 2023). However, it is important to note that since the output from these generative models does not directly influence the forecasting process, there are no guarantees that the resulting dynamics will remain continuous over time. Hua et al. (2024) explore the possibility to incorporate prior information in MLWP diffusion forecasting, by guidance from existing NWP forecasts or climatology, but they do not consider ensemble forecasting.

**Temporal resolution.** Unlike in NWP, where stability conditions dictate the timestep, MLWP models are free to predict at any temporal resolution. Still, the most common approach is to learn to forecast a single short timestep (6h) and iterate this process until the desired lead time (Lam et al., 2023; Chen et al., 2023b). Although intuitive, this process can lead to error accumulation and is impossible to parallelize due to its sequential nature (Bi et al., 2023). Multi-step losses have been shown to reduce the error accumulation for deterministic (Lam et al., 2023) and latent-variable based models (Oskarsson et al., 2024), but are not trivially implemented in diffusion models. Taking longer timesteps (24h) has been shown to give better results (Couairon et al., 2024; Bi et al., 2023), but comes at the loss of temporal resolution. Bi et al. (2023) resolves this by training multiple models to forecast different lead times, which are then combined in different ways to reach the lead times of interest. Nguyen et al. (2023; 2025) use a similar setup, but train a single model taking the forecast lead time as an input. This approach, called *continuous forecasting*, parallelizes the prediction of the fine temporal scales, but has so far only been applied to deterministic models. Other approaches have tried to learn fully time-continuous dynamics by using an ODE to generate forecasts (Verma et al., 2024; Saleem et al., 2024; Rühling Cachay et al., 2023; Kochkov et al., 2024).

**Spatio-temporal forecasting with diffusion models.** Outside of MLWP, diffusion models have also been applied to forecasting other spatio-temporal processes (Yang et al., 2024). Notable examples include turbulent flow simulation (Kohl et al., 2024; Rühling Cachay et al., 2023) and PDE solving (Lippe et al., 2023). Yang & Sommer (2023) apply diffusion models conditioned on the prediction lead time to a specific floating-smoke fluid field, but do not consider ensemble forecasting. Another alternative to autoregressive rollouts is the DYffusion framework (Rühling Cachay et al., 2023), where stochastic interpolation and deterministic forecasting is combined into a diffusion-like model. The method allows for forecasting at arbitrary temporal resolution, but still requires sequential computations for sampling the prediction. To get a probabilistic model, they introduce a layer dropout term in the interpolator that they keep on during inference. This makes the performance sensitive to the dropout rate, which can not be changed without retraining both the interpolator and forecasting networks. Further, the interpolation gives no guarantee of temporal continuity of trajectories, and since its trained with a mean squared error loss, is prone to blurring similar to deterministic forecasting models. DYffusion has been successfully applied to climate modeling (Rühling Cachay et al., 2025), but not weather forecasting.

## 3 BACKGROUND

**Problem statement.** This paper targets the global weather forecasting problem, an initial-value problem with intrinsic uncertainty. Consider a weather state space $\mathcal{X}$ containing the grid of target variables. Given information about previous weather states $X(\Omega) \subset \mathcal{X}$ at times $\Omega \subset (-\infty, 0]$, the aim is to forecast a trajectory $X(\mathcal{T})$ of future weather states $X : (-\infty, \infty) \to \mathcal{X}$ at times $\mathcal{T} \subset (0, T]$ for some time horizon $T$. In particular, the task is to learn and sample from the conditional distribution $p(X(\mathcal{T})|X(\Omega))$.

**Autoregressive forecasting.** To simplify the problem, $\mathcal{T}$ is often chosen as a set of discrete equally spaced times $\{k\delta\}_{k=1}^{N}$ for some timestep $\delta$. By choosing $\Omega = \{-k\delta\}_{k=0}^{M}$ and assuming $M$th order Markovian dynamics, the joint distribution of $X[k] := X(k\delta)$ can be factorized over successive states,

$$p(X[1{:}N]|X[-M{:}0]) = \prod_{k=0}^{N-1} p(X[k{+}1]|X[k{-}M{:}k]), \qquad (1)$$

and the forecasts can be sampled autoregressively. This way, the network only has to learn to sample a single step $p(X[k{+}1]|X[k{-}M{:}k])$.

**Conditional Diffusion Models.** Similarly to Price et al. (2025), we model $p(X[k{+}1]|X[k{-}M{:}k])$ using a conditional diffusion model (Ho et al., 2020; Song et al., 2021; Karras et al., 2022). Diffusion models generate samples by iteratively transforming noise into data. To forecast a future weather state $X[k{+}1]$ given $X[k{-}M{:}k]$ we sample random noise from $p_{\text{noise}}$ and iteratively transform it until it resembles a sample from $p(X[k{+}1]|X[k{-}M{:}k])$. We consider the SDE formulation of diffusion models presented by Karras et al. (2022), but remark that our framework generalizes to any diffusion or flow matching framework based on stochastic differential equations. Sampling can then be done by solving the *probability flow ODE*

$$z(0) = z(1) - \int_0^1 \dot{\sigma}_s \sigma_s S_\theta\left(z(s), \sigma_s; X[k{-}M{:}k]\right) \mathrm{d}s, \quad z(1) \sim p_{\text{noise}} \qquad (2)$$

starting in pure noise and ending in our forecast $z(0) \sim p(X[k{+}1]|X[k{-}M{:}k])$. The ODE-solver can be chosen freely, we employ the second-order Heun's method. Here $S_\theta$ is the neural network trained to match the score function $\nabla_z \log p_s$ through the denoising training objective as presented by Karras et al. (2022). Similar to (Karras et al., 2022), we choose $\sigma_s = s$ and feed the noise level $\sigma$ to $S_\theta$ in each layer as a Fourier embedding. Repeating this process for different noise samples $z(1)$ gives an ensemble of forecasts. For a more detailed description of diffusion models see appendix B.

## 4 CONTINUOUS ENSEMBLE FORECASTING

Autoregressive forecasting models are simple to train but can suffer from error accumulation at long horizons. Consider again the general problem of sampling from $p(X(\mathcal{T})|X(\Omega))$. In *continuous forecasting*, a single-step prediction is given by conditioning on the lead time $t \in \mathcal{T}$ and training a conditional score network $S_\theta(z, \sigma; X(\Omega), t)$ to simulate directly from the marginal distribution $p(X(t)|X(\Omega), t)$. This does not require setting a fixed $\delta$, making the setup more flexible. The lead time $t$ is added to the conditioning arguments for clarity and can be passed to the network in the same way as the noise level $\sigma$. While this allows sampling a distribution of states $X(t)$ at each time $t \in \mathcal{T}$, combining these naively does not result in a trajectory $X(\mathcal{T})$. This is because $X(t)$ are samples from the marginal distributions $p(X(t)|X(\Omega), t)$ and not the joint trajectory distribution $p(X(\mathcal{T})|X(\Omega))$. We propose *Continuous Ensemble Forecasting*, a novel method of combining samples from $p(X(t)|X(\Omega), t)$ into forecast trajectories $X(\mathcal{T})$ without resorting to autoregressive predictions.

The core idea in our method is to control the source of randomness. To sample from $p(X(t)|X(\Omega), t)$, we sample some noise $Z \sim p_{\text{noise}}$ and feed it to an ODE-solver that solves the probability flow ODE in eq. (2), giving us a forecast $X(t)$ for lead time $t$. Since the ODE-solver is deterministic, the randomness is limited to the noise initialization $Z$. If we freeze the noise, the ODE-solver becomes a deterministic map $f_\theta^Z : \mathcal{X}^{|\Omega|} \times \mathcal{T} \to \mathcal{X}$ parameterized by the neural network $S_\theta$. Applying this map to previous states $X(\Omega)$ and a time $t \in \mathcal{T}$ gives a forecast $X(t)$. Repeating this for several $t_1, \ldots, t_N \subset \mathcal{T}$ gives a sequence of forecasts $X(\{t_i\}_{i=1}^{N})$. Extending this to all $t \in \mathcal{T}$, we can construct a trajectory $X(\mathcal{T}) = f_\theta^Z(X(\Omega), \mathcal{T})$. We treat this as a sample from $p(X(\mathcal{T})|X(\Omega))$ and propose to use Algorithm 1 to sample such trajectories.

### 4.1 MATHEMATICAL MOTIVATION

In a deterministic system, the dynamics can be described by a forecasting function $f : \mathcal{X}^{|\Omega|} \times \mathcal{T} \to \mathcal{X}$ that maps previous states $X(\Omega) \in \mathcal{X}^{|\Omega|}$, to future states $X(t)$. While weather is in principle

---

**Algorithm 1** The Continuous Ensemble Forecasting algorithm

---

1: **input:** Initial conditions $x(\Omega)$, times $\{t_i\}_{i=1}^N$, ensemble size $n_{\text{ens}}$, network $S_\theta$
2: **sample** $\{z^j\}_{j=1}^{n_{\text{ens}}} \sim \mathcal{N}(0, \mathbf{I})$
3: **for all** $i \in \{1, \ldots, N\}, j \in \{1, \ldots, n_{\text{ens}}\}$ **do**        ▷ Can be done fully in parallel for all $j$ and $i$
4:     $x_i^j \leftarrow$ PROBABILITY-FLOW-SOLVER$(z^j, t_i; x(\Omega), S_\theta)$
    **return** $\{x_i^j\}_{i=1:N}^{j=1:n_{\text{ens}}}$

---

governed by deterministic equations, its chaotic nature, lack of information, and our inability to resolve the dynamics at sufficiently high spatio-temporal resolution gives it an intrinsic uncertainty. This motivates the formulation of weather as a stochastic dynamical system. In such a system, no function can describe the entire dynamics. At each instance, there might be a range of functions $f^1, f^2, \ldots$ that all describe some possible evolution, some more likely than others. If we consider the space $\mathcal{F}$ of all such functions, we can formalize this by defining a probability density $\mu$ over this space, describing the likelihood of each function. To create an ensemble of possible functions, we sample several $f^1, \ldots, f^N$, from $\mu$. Given previous states $X(\Omega)$, evaluating each function gives an ensemble of possible trajectories $X^i(\mathcal{T}) = f^i(X(\Omega), \mathcal{T})$.

The key insight in motivating our method is the identification of the latent noise space $(\mathcal{X}, p_{\text{noise}})$ with the solution space $(\mathcal{F}, \mu)$. In our method, the sampling algorithm can be represented by a parameterized function $f_\theta^Z$, where we have frozen the noise $Z \sim p_{\text{noise}}$. Under the regularity conditions specified in appendix E, this function is uniquely defined by $\theta$ and the ODE-solver for any given $Z$. Thus, as illustrated in fig. 2, our setting mirrors the theoretical setting. Given sufficient data and model capacity, the neural network $S_\theta$ matches the score function $S$. Consequently, the distribution of the functions $f_\theta^Z$ should mirror that of the solutions $f^i$. Since $f^i$ describes a possible evolution of weather, it has to be continuous as a function of time. To ensure that the generated trajectories are also continuous in $t$, regularity conditions need to be imposed on $S_\theta$ to ensure that it depends smoothly on the lead time. In appendix E, we provide sufficient conditions and a proof of this property. We also show empirically in sec. 5.1 that this is satisfied in practice.

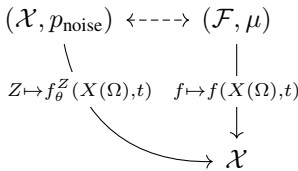

Figure 2: Diagram depicting the connection between the latent noise space and the solution space.

## 4.2 AUTOCORRELATED NOISE

As mentioned above, under regularity conditions on $S_\theta$, we expect the solution map $f_\theta^Z(X(\Omega), t)$ to be continuous in $t$. A shortcoming of this approach, however, is that it also constrains the trajectories to be conditionally deterministic, conditionally on the solution $f_\theta^Z(X(\Omega), t^\star)$ at any fixed time point $t^\star \in \mathcal{T}$. Specifically, consider the conditional distribution $p(X(t) \mid X(t^\star), X(\Omega), t)$ for $t, t^\star \in \mathcal{T}$ under the proposed model. Conditionally on $X(t^\star) = f_\theta^Z(X(\Omega), t^\star)$ we can, conceptually, invert the ODE which generated $X(t^\star)$ to recover the driving noise $Z$. Now, if $X(t) = f_\theta^Z(X(\Omega), t)$ is generated using the exact same noise variable, we find that $p(X(t) \mid X(t^\star), X(\Omega), t)$ is a Dirac point mass concentrated at $f_\theta^Z(X(\Omega), t)$. This violates the assumed stochasticity of the dynamical process that we are modelling.

To address this shortcoming, a simple extension of the proposed model is to replace the fixed $Z$ with a stochastic process $Z(\mathcal{T})$ with continuous sample trajectories. The stochastic process is chosen to be stationary with marginal distribution $Z(t) \sim p_{\text{noise}}, \forall t \in \mathcal{T}$. This ensures that the generative model at any fixed $t$ is probabilistically equivalent with the fixed noise setting. Specifically, no changes to the training algorithm are needed since the training is based solely on time marginals. Allowing temporal stochasticity in the driving noise process results in a non-deterministic relationship between the states at different lead times, while we can keep the temporal consistency by ensuring that the driving process is sufficiently autocorrelated. A simple choice is to select $Z(\mathcal{T})$ as an Ornstein–Uhlenbeck (OU) process (Gillespie, 1996), $dZ(t) = -\rho Z(t)dt + \sqrt{2\rho}dW(t)$, for some correlation parameter $\rho > 0$ and $W(t)$ being Brownian motion. This process can be easily simulated from to generate a noise sequence $\{Z(t_i)\}_{i=1}^N$ for lead-times $\{t_i\}_{i=1}^N$, as done in Algorithm 2. The initial noise sample $Z(0) \sim \mathcal{N}(0, \mathbf{I})$ is perturbed according to the OU process by sampling new

noise $\nu \sim \mathcal{N}(0, \mathbf{I})$ in each timestep. The update step in line 5 comes from the choice of an OU process for $Z(t)$ and corresponds to an exact simulation of the process at the specified lead times $\{t_i\}_{i=1}^N$, i.e., it guarantees the correct marginal distributions $z_i = Z(t_i) \sim \mathcal{N}(0, \mathbf{I})$ for all lead times. An alternative to the OU process is to use a Gaussian process with stronger smoothness properties, e.g., with a squared exponential kernel. This could prove useful for formally proving time continuity of the resulting trajectories also under a stochastic noise process, but we leave such an analysis for future work.

---

**Algorithm 2** The Extended Continuous Ensemble Forecasting algorithm

---

1: **input:** Initial conditions $x(\Omega)$, times $\{t_i\}_{i=1}^N$, ensemble size $n_{\text{ens}}$, network $S_\theta$, correlation parameter $\rho$
2: **sample** $\{z_1^j\}_{j=1}^{n_{\text{ens}}} \sim \mathcal{N}(0, \mathbf{I})$
3: **for** $i = 2$ **to** $N$ **do**
4:     **sample** $\{\nu_i^j\}_{j=1}^{n_{\text{ens}}} \sim \mathcal{N}(0, \mathbf{I})$
5:     $z_i^j \leftarrow \exp(-\rho(t_i - t_{i-1}))z_{i-1}^j + \sqrt{1 - \exp(-2\rho(t_i - t_{i-1}))}\nu_i^j$
6: **for all** $i \in \{1, \ldots, N\}, j \in \{1, \ldots, n_{\text{ens}}\}$ **do**          ▷ Can be done fully in parallel for all $j$ and $i$
7:     $x_i^j \leftarrow$ PROBABILITY-FLOW-SOLVER$(z_i^j, t_i; x(\Omega), S_\theta)$
    **return** $\{x_i^j\}_{i=1:N}^{j=1:n_{\text{ens}}}$

---

### 4.3 AUTOREGRESSIVE ROLL-OUTS WITH CONTINUOUS INTERPOLATION

Continuous forecasting is effective for forecasting hours to days but can struggle to forecast longer lead times where the correlation is weaker. Autoregressive forecasting excels at long lead times when used with longer (24h) timesteps (Bi et al., 2023), but comes at the loss of temporal resolution. In our framework, it becomes possible to sample both autoregressive and continuous forecasts with the same model. We propose to leverage this by iterating on a longer timestep and forecasting the intermediate timesteps using Continuous Ensemble Forecasting, as outlined in Alg. 3. We refer to this combined method as Autoregressive Rollouts with Continuous Interpolation (ARCI). Our method limits the error accumulation without sacrificing temporal resolution. This allows for producing forecasts at an arbitrary fine temporal resolution, while retaining the accuracy of the best autoregressive methods throughout the whole forecast. By limiting the number of autoregressive steps, more of the forecast also becomes parallelizable, allowing for rapidly generating forecasts on large compute clusters. Furthermore, we can straightforwardly use different time resolutions during different parts of the forecast trajectories, by for instance forecasting with 1h steps for the first few days and then switching to longer timesteps for long lead times. Note that ARCI can be used with either fixed (with Alg. 1) or stochastic (with Alg. 2) driving noise. However, we emphasize that these two algorithms are probabilistically equivalent for all time marginals, and only differ in the autocorrelation of forecast trajectories.

---

**Algorithm 3** ARCI (Autoregressive roll-outs with continuous interpolation)

---

1: **input:** Initial conditions $x_{-L:0}$, interpolation times $\{t_i\}_{i=1}^N$, ensemble size $n_{\text{ens}}$, autoregressive steps $M$, network $S_\theta$
2: **for** $m = 0$ **to** $M - 1$ **do**
3:     $\{x_{mN+i}^j\}_{i=1:N}^{j=1:n_{\text{ens}}} \leftarrow$ **Alg. 1**$(x_{mN-L:mN}, \{t_i\}_{i=1}^N, n_{\text{ens}}, S_\theta)$     ▷ Also possible to use **Alg. 2**
    **return** $\{x_i^j\}_{i=1:MN}^{j=1:n_{\text{ens}}}$

---

## 5 EXPERIMENTS

**Data.** We evaluate our method on global weather forecasting up to 10 days at 1, 6, and 24 hour timesteps. We use the downsampled ERA5 reanalysis dataset (Hersbach et al., 2020) at 5.625°resolution and 1-hour increments provided by WeatherBench (Rasp et al., 2020). The models

are trained to forecast 5 variables from the ERA5 dataset: geopotential at 500hPa (z500), temperature at 850hPa (t850), ground temperature (t2m) and the ground wind components (u10, v10). The atmospheric fields z500 and t850 offer a comprehensive view of atmospheric dynamics and thermodynamics, while the surface fields t2m and u10, v10 are important for day-to-day activities. We also evaluate the forecast of ground wind speed ws10, computed from the model outputs as ws10 = $\sqrt{\text{u10}^2 + \text{v10}^2}$. This is useful for evaluating how well the methods model cross-variable dependencies. All variables are standardized by subtracting their mean and dividing by their standard deviation. Together with the previous states we also feed the models with static fields. These include the land-sea mask and orography, both rescaled to $[0, 1]$. All models are trained on the period 1979–2015, validated on 2016–2017 and tested on 2018. We consider every hour of each year as forecast initialization times, except for the first 24h and last 10 days in each subset. This guarantees that all times forecasted or conditioned on lie within the specific years.

**Metrics.** We evaluate the skill of the forecasting models by computing the Root Mean Squared Error (**RMSE**) of the ensemble mean. As a probabilistic metric we also consider Continuous Ranked Probability Score (**CRPS**) (Gneiting & Raftery, 2007), which measures how well the the predicted marginal distributions capture the ground truth. We also evaluate the Spread/Skill-Ratio (**SSR**), which is a common measure of calibration for ensemble forecasts. For a model with well calibrated uncertainty estimates the SSR should be close to 1 (Fortin et al., 2014). Detailed definitions of all metrics are given in appendix A.

**Models.** We propose to use the ARCI model described in algorithm 3 referred to as **ARCI-24/6h**. We train it to forecast $t \in \{6, 12, 18, 24\}$ (hours) and roll it out autoregressively with 24h steps, hence the name. Training is done on a 40GB NVIDIA A100 GPU and takes roughly 2 days. We emphasize again that using fixed, correlated or uncorrelated noise results in probabilistically equivalent forecasts for all time marginals, and only differ in the autocorrelation of forecast trajectories. Hence the choice of algorithm inside ARCI does not matter, and for all metrics below that are computed for specific lead times we only report results for one version of the algorithm. We return to the difference between Alg. 1 and Alg. 2 when studying the temporal difference below.

To evaluate the effectiveness of our approach, we compare it to other MLWP baselines. **Deterministic** is a deterministic model trained using MSE-loss on a single 6 hour timestep, and unrolled up to 10 days. **AR-{6, 24}h** are diffusion models trained only on forecasting a single fixed $\delta = \{6, 24\}$h ahead, and then autoregressively unrolled up to 10 days. This is the exact forecasting setup of (Price et al., 2025) and the AR- models can thus be seen as a reimplementation of GenCast with a U-Net architecture. Since we are interested in forecasts with high temporal resolution, the AR-24h model operating at a coarser temporal resolution represents an upper limit on performance for long rollouts rather than a competing model **CI-6h** is a diffusion model performing continuous forecasting conditioned on a specific lead time. It is trained on uniformly sampled lead times from $\{k\delta\}_{k=1}^{40}$, with $\delta = 6$h. This is the method proposed in alg. 1. Sampling a 10-day forecast with 6h resolution for a single member from AR-6h takes 32 seconds, but by parallelizing the 6h timesteps in ARCI-24/6h this reduces to 8 seconds.

To compare against another family of ensemble forecasting models from the literature we retrain the **Graph-EFM** model (Oskarsson et al., 2024) on our exact data setup. Graph-EFM is a graph-based latent-variable model that produces forecasts by 6h iterative rollout steps. We also present new results from the **DYffusion** model Rühling Cachay et al. (2023), trained on ERA5 data at 1h resolution. All models except Graph-EFM use the same U-net with 3.5M parameters. The architecture is based on the U-net in Karras et al. (2022) and presented in appendix B. Due to its specific forward-backward process, DYffusion only supports conditioning on the current timestep, $\Omega = \{0\}$. For all other models we condition on the two previous timesteps $\Omega = \{0, -\delta\}$.

## 5.1 RESULTS

**Qualitative results.** Figure 3 shows an example forecast from ARCI-24/6h for temperature at 850 hPa (t850) at 10 days lead time. The forecasts are rich in detail, resembling the true state more than the ensemble mean. Examples of other variables are given in appendix D.

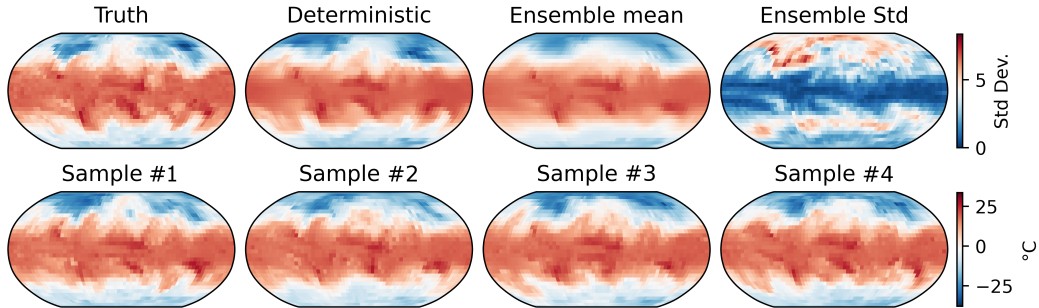

Figure 3: Example forecasts for temperature at 850hPa (t850) at lead time 10 days. The forecasts are generated using ARCI-24/6h except for Deterministic which is sampled using the deterministic model. The bottom row shows 4 ensemble members, randomly chosen out of the 50.

Table 1: Selection of results for 5 and 10 day forecasting using models with 6h resolution for geopotential at 500 hPa (z500) and temperature at 850 hPa (t850). For RMSE and CRPS, lower values are better, and SSR should be close to 1. The best values are marked with **bold** and second best underlined. The AR-24h model is included for reference, but is not considered for best model since it operates at a coarser 24h resolution.

| Variable | Model | Lead time 5 days | | | Lead time 10 days | | |
|---|---|---|---|---|---|---|---|
| | | RMSE | CRPS | SSR | RMSE | CRPS | SSR |
| z500 | Deterministic | 766.7 | 483.9 | - | 1042 | 661.5 | - |
| | Graph-EFM | 699.1 | 317.5 | **1.13** | 817.1 | 373.6 | 1.1 |
| | AR-6h | 602.3 | 287.8 | 0.75 | 811.8 | 391.9 | 0.88 |
| | CI-6h | 707.8 | 321.2 | 0.59 | 885.7 | 406.6 | 0.6 |
| | ARCI-24/6h | **560.9** | **256.7** | 0.86 | **765.6** | **355.2** | **0.93** |
| | AR-24h | 544.2 | 242.7 | 0.84 | 750.6 | 335.2 | 0.94 |
| t850 | Deterministic | 3.48 | 2.36 | - | 4.54 | 3.17 | - |
| | Graph-EFM | 3.12 | 1.56 | **1.11** | 3.51 | 1.77 | 1.12 |
| | AR-6h | 2.72 | 1.34 | 0.82 | 3.39 | 1.69 | 0.92 |
| | CI-6h | 3.06 | 1.5 | 0.74 | 3.68 | 1.85 | 0.71 |
| | ARCI-24/6h | **2.6** | **1.27** | 0.9 | **3.29** | **1.63** | **0.95** |
| | AR-24h | 2.55 | 1.24 | 0.89 | 3.25 | 1.6 | 0.96 |

**Quantitative results.** Table 1 and figure 4a show metrics for a selection of lead times and variables. Scores for the remaining variables are listed in appendix C. For probabilistic models on 6h resolution, we sample 50 ensemble members at each initialization time. All probabilistic models show a clear improvement over the deterministic model. CI-6h performs well on short-term forecasting but struggles at longer horizons. This is likely due to the challenge of learning any useful relationships between initial states and later lead times, which are weakly correlated. ARCI-24+6h outperforms all models at 6h resolution, including the external baseline Graph-EFM and the GenCast setup AR-6h, and matches AR-24h in almost all scores. All diffusion-based models have SSR < 1, indicating some systematic underdispersion. In tables 6, 7 in appendix C we present error bars calculated for the ARCI-24/6h model, which shows that the model is robust to network initialization.

**High Temporal Resolution.** Our ARCI method allows for producing forecasts at high temporal resolution while retaining the accuracy of methods taking longer autoregressive steps. We here demonstrate this by producing hourly forecasts. Figure 4b shows the scores of a selection of models for t850 for 10-day forecasts at 1h resolution. Since the 1h forecasts requires more computational effort, we use only 10 ensemble members instead of 50. The AR-1h model has the same GenCast-setup as AR-6h but on 1h resolution and ARCI-24/1h is trained in the same way as ARCI-24/6h but at 1h resolution and with $\Omega = \{0, -24\}$. The autoregressive AR-1h model performs much worse

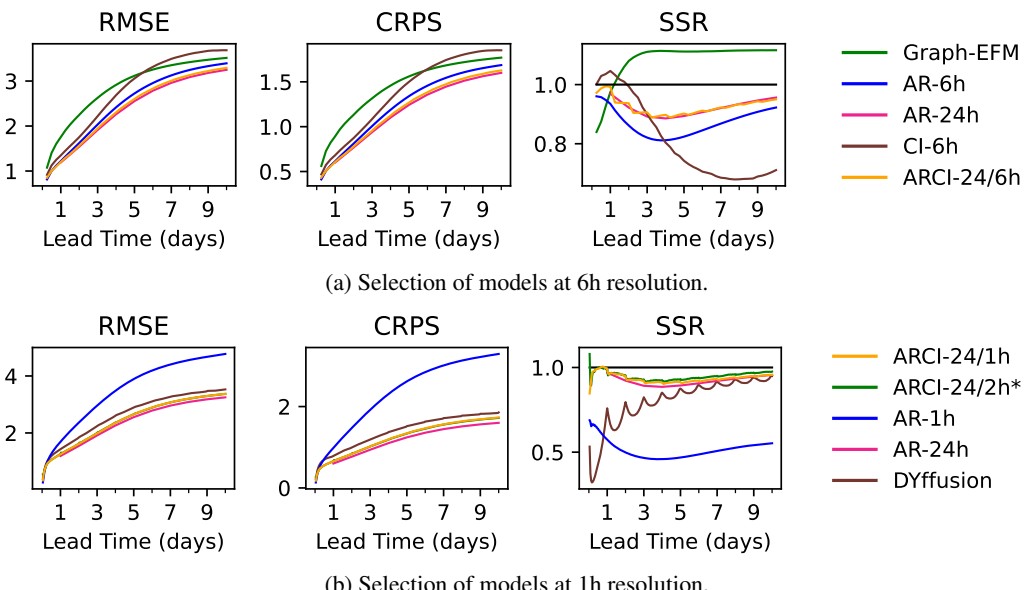

(a) Selection of models at 6h resolution.

(b) Selection of models at 1h resolution.

Figure 4: RMSE, CRPS and SSR for temperature at 850 hPa (t850).

than on 6 or 24 hours. The continuous model, however, does not lose performance by increasing the temporal resolution, making the 1h timestep forecasts as skillful as the 24 hour ones. Although DYffusion achieves much better results than AR-1h, it is beaten by ARCI-24/1h in all variables and lead times.

**Forecast continuity.** By correlating the noise across timesteps, our continuous ensemble forecasting method can produce temporally consistent ensemble trajectories without resorting to autoregressive predictions. Here we give empirical justification for this claim. Since autoregressively sampled forecast trajectories are necessarily continuous, there is no standard way of measuring the continuity of a forecast. We propose using the mean temporal difference $\Delta X = |X(t) - X(t-1)|$ as a measure of forecast continuity. Figure 5 shows $\Delta X$ for a CI-1h continuous model trained up to 24 hours with 1-hour timesteps. Compared to using different noise at each step ($\rho \to \infty$), the temporal difference of our model ($\rho = 0$) stays close to the

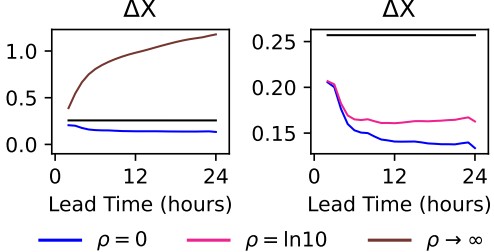

Figure 5: Temporal difference for temperature on 850 hPa (t850) for different values of $\rho$ in algorithm 2. Choosing $\rho = 0$ fixes the noise, $\rho = \ln 10$ allows it to vary and $\rho \to \infty$ gives completely uncorrelated noise. The black line refers to the temporal difference of the data.

temporal difference of the data. This supports our claim that continuous ensemble forecasting produces continuous trajectories. When the noise is fixed ($\rho = 0$) the temporal difference decreases with lead time, corresponding to predictions with smaller temporal variations. Letting the noise vary with noise factor ($\rho = \ln 10$) as in alg. 2 stops this from happening. The bias between $\Delta X$ of the data and our model is likely due to the model producing slightly blurrier forecasts, making the differences smaller.

**Continuous Time Forecasting.** One of the key advantages of continuous ensemble forecasting is its ability to generate forecasts at arbitrarily fine temporal resolutions. To test this, we train a model at a coarser temporal resolution and evaluate it at a higher resolution. Specifically, we consider the ARCI-24/2h* model, which is trained on lead times spaced 2 hours apart but evaluated at every 1-hour timestep. To ensure a fair comparison with ARCI-24/1h, ARCI-24/2h* is trained with $\Omega = \{0, -24\}$, providing access to the same information at each timestep. As shown in fig. 4b,

ARCI-24/2h* performs similarly to ARCI-24/1h, demonstrating its ability to generalize beyond the temporal resolution seen during training.

For highly time-dependent fields such as t2m, ARCI-24/2h* performs worse at the first forecast in each iteration (1h, 25h,...), as seen in figure 10 in appendix C. For lead-times $t$ not considered during training, the model has trained on forecasting $t-1$ and $t+1$, thus only having to interpolate to $t$. However, since we do not train on forecasting 0h, the model instead has to extrapolate to $t = 1h$ what was learned for 2h forecasts. This issue could possibly be fixed by letting the network also train on 0h forecasts.

An alternative to directly producing forecasts at a fine temporal resolution would be to linearly interpolate the forecasts sampled using an autoregressive model. In fig. 8 in appendix C we show that linearly interpolated forecasts behave much worse than the ARCI model on both 1 and 6-hour resolution.

## 6 CONCLUSION

We present Continuous Ensemble Forecasting, a novel framework for probabilistic MLWP that increases the efficiency, accuracy, and flexibility of weather forecasts at high temporal resolution. When combined with autoregressive prediction, our ARCI method can produce accurate 10-day global ensemble weather forecasts with a 1-hour resolution. It achieves the same accuracy as a purely autoregressive model with 24-hour steps and surpasses models like GenCast and DYffusion when reimplemented using the same neural network architecture. With this work, we hope to show that the possibilities with generative modeling for spatio-temporal predictions are still largely unexplored and a fruitful area of research.

**Limitations.** While our proposed framework achieves good results on 5.625°Weatherbench (Rasp et al., 2020) data, we have yet to show that the method scales to problems with higher spatial resolution. Additionally, as the lead time increases, the correlation between initial and future states becomes weaker, limiting the application of continuous forecasting. While our method parallelizes more of the sampling than previous autoregressive models, sampling from the diffusion model still requires many sequential forward passes through the network. Sampling is thus still slower than for latent variable models, but the predicted distribution more accurate.

**Future work.** One interesting direction for future work is a further investigation of autocorrelated noise, in particular, how the choice of stochastic process can aid in producing continuous trajectories with a stationary temporal difference. This includes correlating the noise in the autoregressive steps with the continuous steps, which could help ease the transition between them. Another idea is to take the direction of DYffusion (Rühling Cachay et al., 2023) and directly adjust the diffusion objective to better suit temporal data. While we have demonstrated continuous ensemble forecasting for weather, the idea is generally applicable and it would also be of interest to apply it to other spatio-temporal forecasting problems.

## ACKNOWLEDGMENTS

This work was financially supported by the Wallenberg AI, Autonomous Systems and Software Program (WASP) funded by the Knut and Alice Wallenberg Foundation, the Swedish Research Council (project no: 2020-04122, 2024-05011), and the Excellence Center at Linköping–Lund in Information Technology (ELLIIT). Our computations were enabled by the Berzelius resource at the National Supercomputer Centre, provided by the Knut and Alice Wallenberg Foundation.

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

## A  METRICS

We consider the following evaluation metrics used to assess the probabilistic forecasts produced by the diffusion model. Metrics from meteorology and general uncertainty quantification, such as RMSE, Spread/Skill ratio (SSR), and Continuous Ranked Probability Score (CRPS) are employed to measure the effectiveness and reliability of the model outputs. All of our metrics are weighted by latitude-dependent weights. For a particular variable and lead time,

- $x_{i,n}^k$ represents the value of the $k$-th ensemble member at initialization time indexed by $n = 1 \ldots N$ for grid cell in the latitude and longitude grid indexed by $i \in I$.

- $y_{i,n}$ denotes the corresponding ground truth.

- $\bar{x}_{i,n}$ denotes the ensemble mean, defined by $\bar{x}_{i,n} = \frac{1}{n_{\text{ens}}} \sum_{k=1}^{n_{\text{ens}}} x_{i,n}^k$.

- $a_i$ denotes the area of the latitude-longitude grid cell, which varies by latitude and is normalized to unit mean over the grid (Rasp et al., 2020).

**RMSE** or skill measures the accuracy of the forecast. Following Rasp et al. (2020) we define the RMSE as the mean square root of the ensemble mean:

$$\text{RMSE} := \frac{1}{N} \sum_{n=1}^{N} \sqrt{\frac{1}{|I|} \sum_{i \in I} a_i (y_{i,n} - \bar{x}_{i,n})^2}. \tag{3}$$

In the case of deterministic predictions, the ensemble mean is taken as the deterministic prediction.

**Spread** represents the variability within the ensemble and is calculated as the root mean square of the ensemble variance:

$$\text{Spread} := \frac{1}{N} \sum_{n=1}^{N} \sqrt{\frac{1}{|I|} \sum_{i \in I} \frac{1}{n_{\text{ens}} - 1} \sum_{k=1}^{n_{\text{ens}}} a_i (x_{i,n}^k - \bar{x}_{i,n})^2}. \tag{4}$$

Ideally, the forecast achieves a balance where skill and spread are proportional, leading to an optimal spread/skill ratio (SSR) close to 1, indicating effective uncertainty estimation:

$$\text{SSR} := \sqrt{\frac{n_{\text{ens}} + 1}{n_{\text{ens}}}} \frac{\text{Spread}}{\text{RMSE}}. \tag{5}$$

**Continuous Ranked Probability Score (CRPS)** (Gneiting & Raftery, 2007) measures the accuracy of probabilistic forecasts by comparing the cumulative distribution functions (CDFs) of the predicted and observed values. It integrates the squared differences between these CDFs, providing a single score that penalizes differences in location, spread, and shape of the distributions. An estimator of the CRPS is given by:

$$\text{CRPS} := \frac{1}{N} \sum_{n=1}^{N} \frac{1}{|I|} \sum_{i \in I} a_i \left( \frac{1}{n_{\text{ens}}} \sum_{k=1}^{n_{\text{ens}}} |x_{i,n}^k - y_{i,n}| - \frac{1}{2 n_{\text{ens}}^2} \sum_{k=1}^{n_{\text{ens}}} \sum_{k'=1}^{n_{\text{ens}}} |x_{i,n}^k - x_{i,n}^{k'}| \right).$$

**Temporal Difference** measures the mean absolute difference between states at consecutive times. It's used to measure the continuity of a forecast. For forecasts $x_{i,n}^k, \hat{x}_{i,n}^k$ at consecutive lead times, it is given by:

$$\Delta X := \frac{1}{N} \sum_{n=1}^{N} \frac{1}{n_{\text{ens}}} \sum_{k=1}^{n_{\text{ens}}} \frac{1}{|I|} \sum_{i \in I} a_i |x_{i,n}^k - \hat{x}_{i,n}^k|. \tag{6}$$

## B  MODEL

Score-based generative models uses a parameterized score $S_\theta$ to sample from the target distribution. In practice, it turns out to be easier to learn a denoising network $D_\theta$ using the denoising score matching objective Ho et al. (2020). By a result shown in Vincent (2011), the score can then be retrieved using $S_\theta = (D_\theta - z)/\sigma^2$.

Table 2: Scaling functions.

| | | |
|---|---|---|
| Skip scaling | $c_{\text{skip}}(\sigma)$ | $\sigma_{\text{data}}^2/(\sigma^2 + \sigma_{\text{data}}^2)$ |
| Output scaling | $c_{\text{out}}(\sigma)$ | $\sigma \cdot \sigma_{\text{data}}/\sqrt{\sigma^2 + \sigma_{\text{data}}^2}$ |
| Input scaling | $c_{\text{in}}(\sigma)$ | $1/\sqrt{\sigma^2 + \sigma_{\text{data}}^2}$ |
| Noise scaling | $c_{\text{noise}}(\sigma)$ | $\frac{1}{4}\ln(\sigma)$ |

**Preconditioning**  The sampling process is based on the denoising neural network $D_\theta$ that takes a noisy residual and tries to denoise it. To help in this, it is also given the noise level $\sigma$, the previous state $X(\Omega)$ and the lead time $t$. To simplify learning, $D_\theta$ is parameterized by a different network $F_\theta$ defined by

$$D_\theta(z, \sigma; X(\Omega), t) = c_{\text{skip}}(\sigma) \cdot z + c_{\text{out}}(\sigma) \cdot F_\theta\left(c_{\text{in}}(\sigma) \cdot z, c_{\text{noise}}(\sigma); X(\Omega), t\right),$$

where $c_{\text{skip}}$, $c_{\text{out}}$, $c_{\text{in}}$ and $c_{\text{noise}}$ are scaling functions taken from (Karras et al., 2022) defined in Tab. 2. These scaling functions $c_{\text{skip}}$, $c_{\text{out}}$, $c_{\text{in}}$ and $c_{\text{noise}}$, are specifically chosen to handle the influence of the noise level within the network, allowing $D_\theta$ to adapt dynamically to different noise intensities without the need for adjusting the scale of $\sigma$ externally. Consequently, for consistency with the normalization of the data where $\sigma_{\text{data}}$ is set to 1, the lead time $t$ is also scaled to fit within the range $[0, 1]$. This normalization ensures that the network inputs are uniformly scaled, enhancing the efficiency and effectiveness of the denoising process.

**Conditioning**  To condition on the initial conditions $X(\Omega)$ and static fields, these are concatenated along the channel dimension with the input to the denoiser, increasing the dimension of the input. To condition on the noise level $\sigma$ and lead time $t$, we use Fourier embedding as specified in (Karras et al., 2022). Fourier embedding captures periodic patterns in noise and time, enhancing the model's ability to handle complex time-series dependencies effectively. They work by transforming the time/noise into a vector of sine/cosine features at 32 frequencies with period 16. These vectors are added and then passed through two fully connected layers with SiLU activation to obtain a 128-dimensional encoding.

**Architecture**  The backbone of the diffusion model is a U-Net architecture. Our model is based on the one used in (Karras et al., 2022), reconfigured for our purposes with 32 filters as the base multiplier. It is built up by blocks configured as in fig. 6. The blocks consist of two convolutional layers and are constructed as in fig. 7. If the block is a down-/up-sample or if the number of input filters is different from the number of output filters, there is an additional skip layer from the input to the output. The blocks at $16 \times 32$ resolution additionally has attention with a single head. The time/noise embedding is fed directly into each block and not passed through the network. Unlike the network in Karras et al. (2022), our convolutions uses zero padding at the poles and periodic padding at the left/right edges. This periodic padding ensures periodicity over longitudes. The model has 3.5M parameters.

**Sampling**  To generate forecasts using our diffusion model, we solve the probability flow ODE as defined in (Karras et al., 2022)

$$\mathrm{d}z = -\dot{\sigma}_s \sigma_s \nabla_z \log p_s(z)\,\mathrm{d}s. \tag{7}$$

We employ the second-order Heun's method, a deterministic ODE solver, as outlined in Algorithm 4. For the noise parameters, we define the noise level function as $\sigma_s = s$. Additionally, we set a noise level schedule to lower the noise during sampling from $\sigma_{\text{max}}$ to $\sigma_{\text{min}}$ over $N$ steps:

$$s_i = \left(\sigma_{\text{max}}^{\frac{1}{\rho}} + \frac{i}{N-1}\left(\sigma_{\text{min}}^{\frac{1}{\rho}} - \sigma_{\text{max}}^{\frac{1}{\rho}}\right)\right)^\rho, \quad i \in \{0, \dots, N-1\}.$$

The relevant parameters for training and sampling are given in tab. 3.

**Training**  The dataset is partitioned into three subsets: training, validation, and testing. The training subset is used for model training, the validation subset for evaluating generalization, and the testing subset to determine final accuracy. The diffusion model is trained using the following training objective

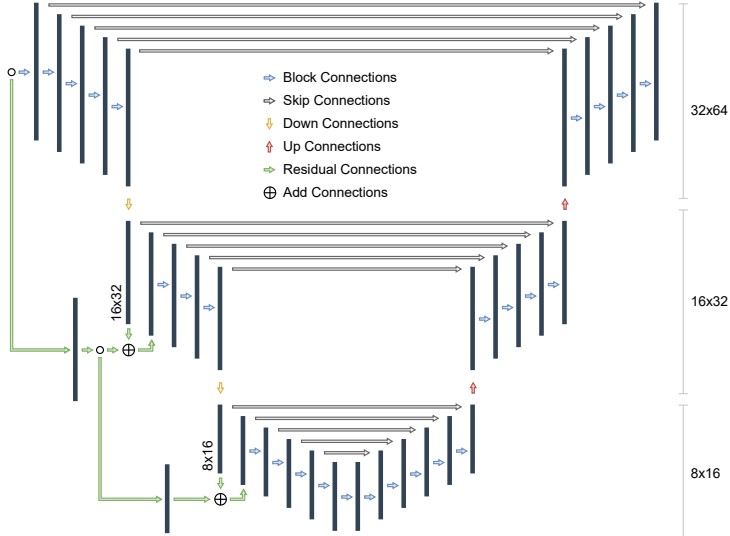

Figure 6: Overview of the U-Net Architecture, detailing layer configurations and the flow of information through convolutional blocks and skip connections.

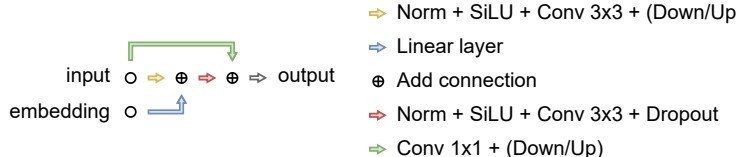

Figure 7: Construction of a Diffusion Model Block, showing the sequence of operations and the integration of embeddings with add connections.

---

**Algorithm 4** Deterministic sampling using Heun's 2$^{\text{nd}}$ order method.

1: **procedure** HEUNSAMPLER($D_\theta(z; \sigma, X(\Omega), t), s_{i \in \{0,...,N\}}, Z$)
2: $z_0 \leftarrow \sigma^2(s_0) \cdot Z$          ▷ Generate initial sample at $s_0$
3: **for** $i = 0$ **to** $N - 1$ **do**          ▷ Solve ODE over $N$ timesteps
4:      $\boldsymbol{d}_i \leftarrow \frac{\dot{\sigma}_{s_i}}{\sigma_{s_i}}(z_i - D_\theta(z_i; \sigma_{s_i}, X(\Omega), t))$          ▷ Evaluate $dz/ds$ at $s_i$
5:      $z_{i+1} \leftarrow z_i + (s_{i+1} - s_i)\boldsymbol{d}_i$          ▷ Take Euler step from $s_i$ to $s_{i+1}$
6:      **if** $s_{i+1} \neq 0$ **then**          ▷ Apply 2$^{\text{nd}}$ order correction unless $\sigma$ goes to zero
7:          $\boldsymbol{d}'_i \leftarrow \frac{\dot{\sigma}_{s_{i+1}}}{\sigma_{s_{i+1}}}(z_{i+1} - D_\theta(z_{i+1}; \sigma_{s_{i+1}}, X(\Omega), t))$          ▷ Evaluate $dz/ds$ at $s_{i+1}$
8:          $z_{i+1} \leftarrow z_i + \frac{1}{2}(s_{i+1} - s_i)(\boldsymbol{d}_i + \boldsymbol{d}'_i)$          ▷ Explicit trapezoidal rule at $s_{i+1}$
     **return** $z_N$          ▷ Return noise-free sample at $s_N$

---

Table 3: Parameters used for sampling and training.

| Name | Notation | Value, sampling | Value, training |
|---|---|---|---|
| Maximum noise level | $\sigma_{\text{max}}$ | 80 | 88 |
| Minimum noise level | $\sigma_{\text{min}}$ | 0.03 | 0.02 |
| Shape of noise distribution | $\rho$ | 7 | 7 |
| Number of noise levels | $N$ | 20 | |

Table 4: Optimizer Hyperparameters.

| Optimizer hyperparameters | |
|---|---|
| Optimiser | AdamW (Loshchilov & Hutter, 2017a) |
| Initialization | Xavier Uniform (Glorot & Bengio, 2010) |
| LR decay schedule | Cosine (Loshchilov & Hutter, 2017b) |
| Peak LR | 5e-4 |
| Weight decay | 0.1 |
| Warmup steps | 1e3 |
| Epochs | 300 |
| Batch size | 256 |
| Dropout probability | 0.1 |

Table 5: Training schedule for Graph-EFM, using the notation from Oskarsson et al. (2024).

| Epochs | Learning Rate | Unrolling steps | $\lambda_{\text{KL}}$ | $\lambda_{\text{CRPS}}$ |
|---|---|---|---|---|
| 20 | $10^{-3}$ | 1 | 0 | 0 |
| 75 | $10^{-3}$ | 1 | 0.1 | 0 |
| 20 | $10^{-4}$ | 4 | 0.1 | 0 |
| 8 | $10^{-4}$ | 8 | 0.1 | $10^5$ |

$$\mathbb{E}_{t\sim p^t}\mathbb{E}_{\sigma\sim p_\sigma}\mathbb{E}_{(X(\Omega),X(t))\sim p_{\text{data}}}\mathbb{E}_{\epsilon|\sigma\sim\mathcal{N}(0,\sigma^2\mathbf{I})}\left[\frac{1}{\sigma^2}\sum_i\sum_j\frac{a_i}{s_j(t)}\frac{1}{|I||J|}\left(\hat{X}(t)_{i,j}-X(t)_{i,j}\right)^2\right].$$

with $\hat{X}(t) = D_\theta(X(t) + \epsilon; \sigma, X(\Omega), t)$ and $J$ being the set of variables. Here, $p^t$ represents a uniform distribution over the lead times. We have also included a scaling term $s_j(t)^{-1}$ which scales the loss by the precomputed standard deviation $s_j(t)$ based on lead time $t$ for each variable $j \in J$. This normalization process is designed to weigh short and longer times equally. The noise level distribution $p_\sigma$ is chosen to be consistent with the sampling noise level described above. Specifically, its inverse CDF is:

$$F^{-1}(u) = \left(\sigma_{\max}^{\frac{1}{\rho}} + u\left(\sigma_{\min}^{\frac{1}{\rho}} - \sigma_{\max}^{\frac{1}{\rho}}\right)\right)^\rho,$$

and we sample from it by drawing $u \sim U[0, 1]$. The training process is executed in Pytorch, with setup and parameters detailed in Tab. 4.

**Graph-EFM Baseline**   For the Graph-EFM baseline we use the same data setup as for the other models. Since we are working on a coarser resolution than Oskarsson et al. (2024) some adaptation was necessary to the exact graph structure used in the model. We construct the graph by splitting a global icosahedron 2 times, resulting in 3 hierarchical graph levels. The training follows the same schedule as in Oskarsson et al. (2024), with pre-training on single step 6h prediction and fine-tuning on rollouts. Details of the training schedule are given in tab. 5.

**DYffusion Baseline**   We have adapted the DYffusion model to work with weather forecasting using ERA5 data. This application was not considered in the original DYffusion papers which focused on sea-surface forecasting Rühling Cachay et al. (2023) and climate modeling Rühling Cachay et al. (2025). To ensure a fair evaluation, we used the same backbone U-Net as our diffusion models, and a lat-lon weighted mean-squared-error loss. We have trained it with a horizon of 24h, with 1h timesteps, making it directly comparable to ARCI-24/1h. The dropout rate was set to 0.2 and we used no artificial diffusion steps. For the training hyperparameters we used values similar to those proposed in the original paper.

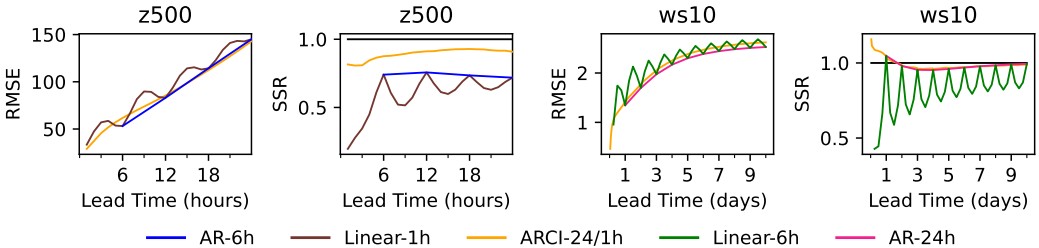

Figure 8: RMSE and SSR for linear interpolations of geopotential at 500 hPa (`z500`) and ground wind speed (`ws10`). Linear-1h/Linear-6h are linear interpolations of AR-6h/AR-24h.

## C  ADDITIONAL RESULTS

**Interpolation**  An alternative to directly producing forecasts at a fine temporal resolution would be to simply interpolate the forecasts sampled using an autoregressive model. We argue that our ARCI method produces more accurate and realistic predictions than simple linear interpolation. Figure 8 shows the RMSE and Spread/Skill for a linear interpolation of AR-24h, compared to ARCI-24/6h. The interpolated forecasts behave much worse than the ARCI model on the same resolution, dismissing this argument.

Table 6, 7 and show the results on 5 and 10 day forecasting for all variables. The error bars given for ARCI-24/6h is a standard deviation calculated by retraining the model five times and evaluating each model on 10 ensemble members. This shows that the model is robust to network initialization. Figure 9 show the results on forecasting at 6h resolution for all variables. Figure 10 shows the same results on forecasting with 1h resolution. Figure 11 shows the temporal difference for different values of $\rho$.

## D  EXAMPLE FORECASTS

Figures 12–17 show example forecasts for the remaining variables.

Table 6: Results on 5 day forecasting for all variables. For RMSE and CRPS, lower values are better, and SSR should be close to 1.

| Variable | Model | Lead time 5 days | | |
| | | RMSE | CRPS | SSR |
| --- | --- | --- | --- | --- |
| z500 | Deterministic | 766.7 | 483.9 | - |
| | Graph-EFM | 699.1 | 317.5 | 1.13 |
| | AR-6h | 602.3 | 287.8 | 0.75 |
| | AR-24h | 544.2 | 242.7 | 0.84 |
| | CI-6h | 707.8 | 321.2 | 0.59 |
| | ARCI-24/6h | $560.9 \pm 15.6$ | $256.7 \pm 20.2$ | $0.86 \pm 0.015$ |
| t850 | Deterministic | 3.48 | 2.36 | - |
| | Graph-EFM | 3.12 | 1.56 | 1.11 |
| | AR-6h | 2.72 | 1.34 | 0.82 |
| | AR-24h | 2.55 | 1.24 | 0.89 |
| | CI-6h | 3.06 | 1.5 | 0.74 |
| | ARCI-24/6h | $2.6 \pm 0.031$ | $1.27 \pm 0.021$ | $0.9 \pm 0.012$ |
| t2m | Deterministic | 2.71 | 1.72 | - |
| | Graph-EFM | 2.51 | 1.1 | 1.09 |
| | AR-6h | 2.13 | 0.96 | 0.83 |
| | AR-24h | 1.98 | 0.87 | 0.9 |
| | CI-6h | 2.29 | 1.0 | 0.79 |
| | ARCI-24/6h | $2.02 \pm 0.036$ | $0.9 \pm 0.035$ | $0.9 \pm 0.012$ |
| u10 | Deterministic | 4.37 | 2.93 | - |
| | Graph-EFM | 3.81 | 1.93 | 0.97 |
| | AR-6h | 3.47 | 1.71 | 0.86 |
| | AR-24h | 3.32 | 1.62 | 0.92 |
| | CI-6h | 3.87 | 1.91 | 0.77 |
| | ARCI-24/6h | $3.35 \pm 0.032$ | $1.64 \pm 0.019$ | $0.93 \pm 0.007$ |
| v10 | Deterministic | 4.48 | 3.0 | - |
| | Graph-EFM | 3.88 | 1.96 | 0.94 |
| | AR-6h | 3.55 | 1.76 | 0.86 |
| | AR-24h | 3.42 | 1.68 | 0.93 |
| | CI-6h | 4.0 | 1.99 | 0.77 |
| | ARCI-24/6h | $3.45 \pm 0.026$ | $1.69 \pm 0.014$ | $0.94 \pm 0.006$ |
| ws10 | Deterministic | 3.09 | 2.2 | - |
| | Graph-EFM | 2.56 | 1.35 | 1.01 |
| | AR-6h | 2.38 | 1.23 | 0.9 |
| | AR-24h | 2.3 | 1.18 | 0.96 |
| | CI-6h | 2.54 | 1.32 | 0.88 |
| | ARCI-24/6h | $2.31 \pm 0.016$ | $1.19 \pm 0.01$ | $0.96 \pm 0.006$ |

Table 7: Results on 10 day forecasting for all variables. For RMSE and CRPS, lower values are better, and SSR should be close to 1.

| | | Lead time 10 days | | |
|---|---|---|---|---|
| Variable | Model | RMSE | CRPS | SSR |
| z500 | Deterministic | 1042 | 661.5 | - |
| | Graph-EFM | 817.1 | 373.6 | 1.1 |
| | AR-6h | 811.8 | 391.9 | 0.88 |
| | AR-24h | 750.6 | 335.2 | 0.94 |
| | CI-6h | 885.7 | 406.6 | 0.6 |
| | ARCI-24/6h | $765.6 \pm 21.4$ | $355.2 \pm 33.0$ | $0.93 \pm 0.018$ |
| t850 | Deterministic | 4.54 | 3.17 | - |
| | Graph-EFM | 3.51 | 1.77 | 1.12 |
| | AR-6h | 3.39 | 1.69 | 0.92 |
| | AR-24h | 3.25 | 1.6 | 0.96 |
| | CI-6h | 3.68 | 1.85 | 0.71 |
| | ARCI-24/6h | $3.29 \pm 0.05$ | $1.63 \pm 0.041$ | $0.95 \pm 0.007$ |
| t2m | Deterministic | 3.56 | 2.28 | - |
| | Graph-EFM | 2.88 | 1.32 | 1.14 |
| | AR-6h | 2.62 | 1.18 | 0.89 |
| | AR-24h | 2.48 | 1.09 | 0.95 |
| | CI-6h | 2.75 | 1.21 | 0.75 |
| | ARCI-24/6h | $2.51 \pm 0.068$ | $1.11 \pm 0.076$ | $0.94 \pm 0.017$ |
| u10 | Deterministic | 5.14 | 3.53 | - |
| | Graph-EFM | 4.08 | 2.07 | 0.97 |
| | AR-6h | 3.95 | 1.97 | 0.94 |
| | AR-24h | 3.85 | 1.9 | 0.97 |
| | CI-6h | 4.27 | 2.14 | 0.78 |
| | ARCI-24/6h | $3.87 \pm 0.03$ | $1.92 \pm 0.018$ | $0.97 \pm 0.01$ |
| v10 | Deterministic | 5.23 | 3.58 | - |
| | Graph-EFM | 4.11 | 2.08 | 0.93 |
| | AR-6h | 4.02 | 2.0 | 0.96 |
| | AR-24h | 3.96 | 1.96 | 0.99 |
| | CI-6h | 4.39 | 2.2 | 0.8 |
| | ARCI-24/6h | $3.97 \pm 0.018$ | $1.97 \pm 0.011$ | $0.99 \pm 0.01$ |
| ws10 | Deterministic | 3.44 | 2.49 | - |
| | Graph-EFM | 2.65 | 1.4 | 1.01 |
| | AR-6h | 2.57 | 1.34 | 0.96 |
| | AR-24h | 2.53 | 1.31 | 0.99 |
| | CI-6h | 2.68 | 1.4 | 0.89 |
| | ARCI-24/6h | $2.54 \pm 0.011$ | $1.32 \pm 0.008$ | $0.99 \pm 0.008$ |

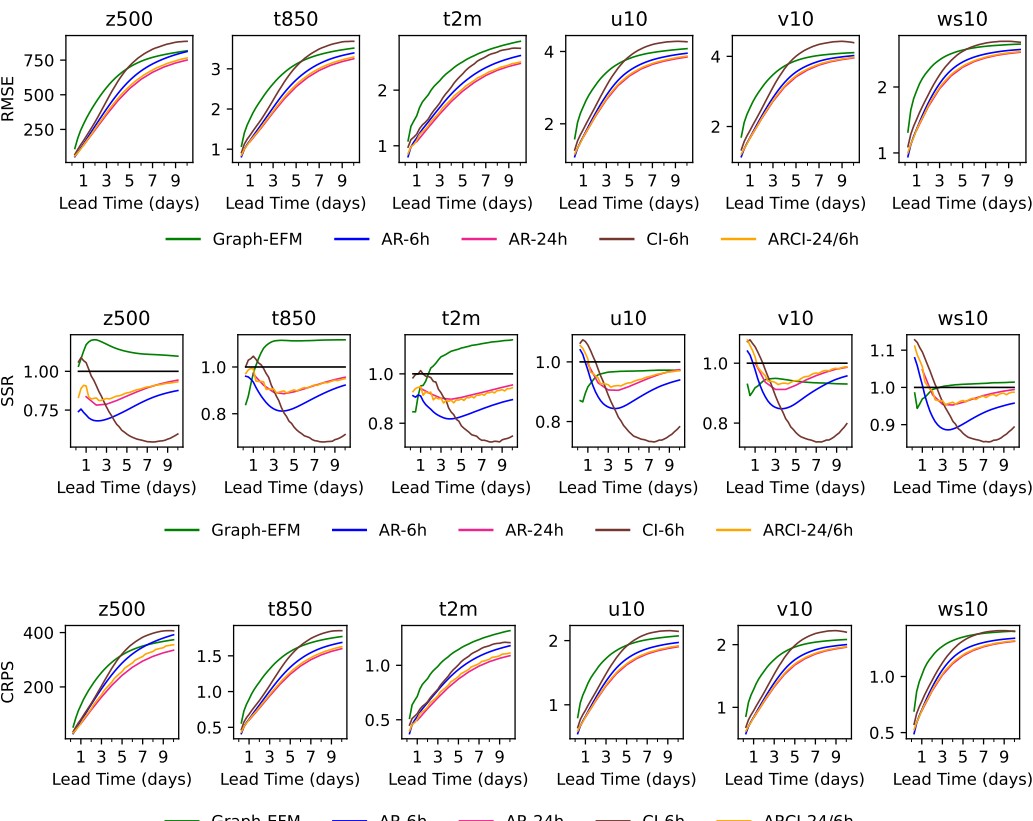

Figure 9: RMSE, SSR, and CRPS for a selection of models at 6h resolution.

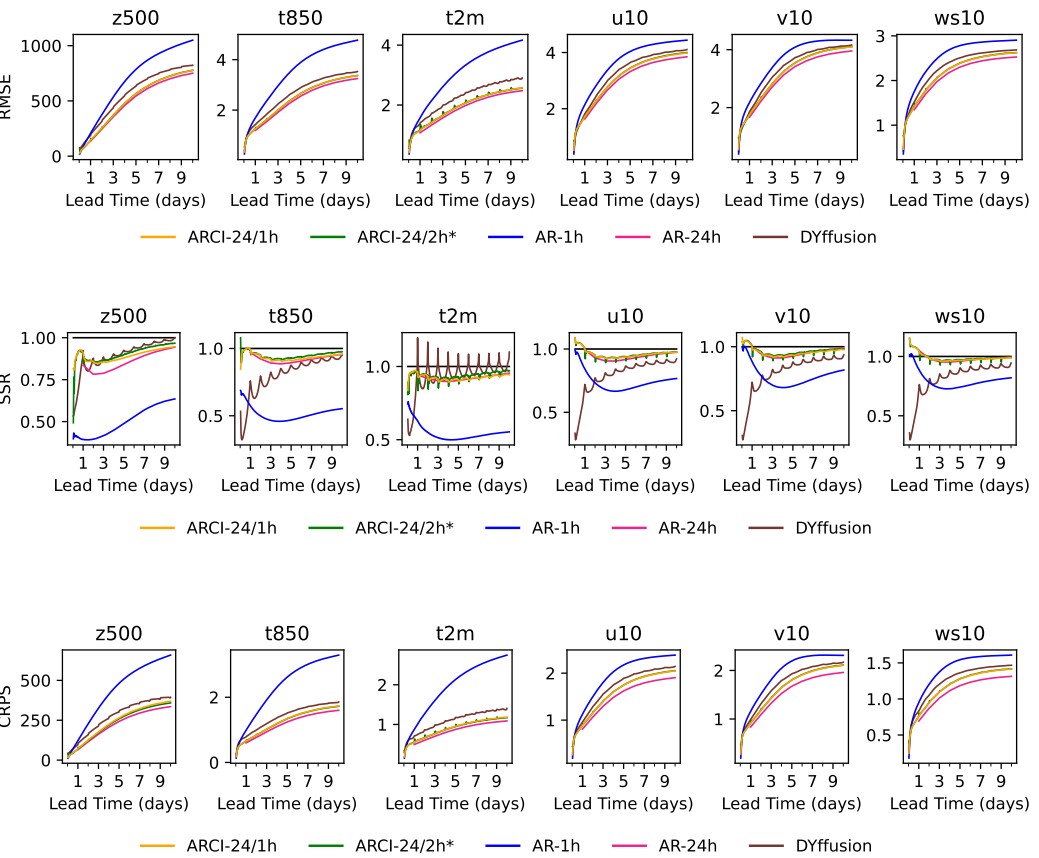

Figure 10: RMSE, SSR and CRPS for a selection of models at 1h resolution.

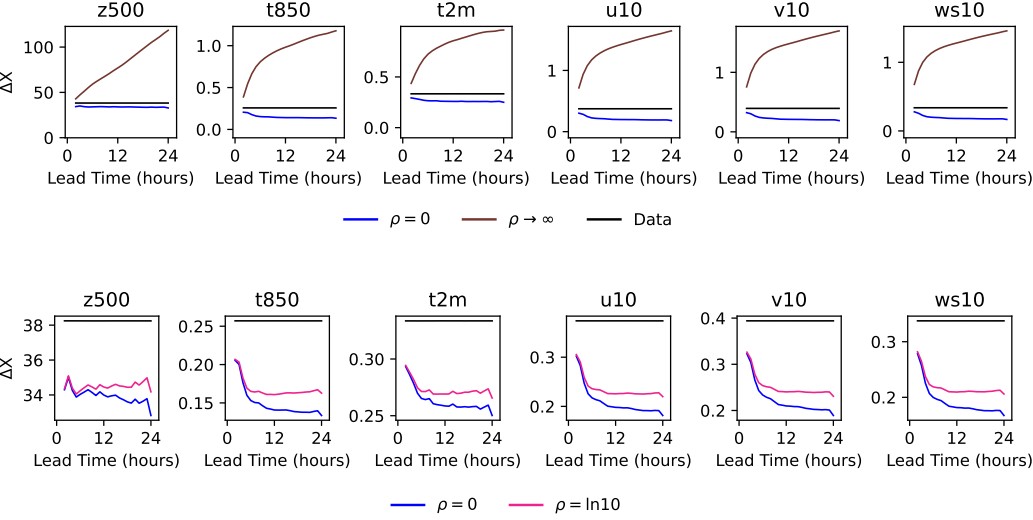

Figure 11: Temporal difference for different values of $\epsilon$ in algorithm 2. Choosing $\rho = 0$ fixes the noise, $\rho = \ln(10)$ allows it to vary slightly and $\rho \to \infty$ gives completely uncorrelated noise. The black line refers to the temporal difference of the data.

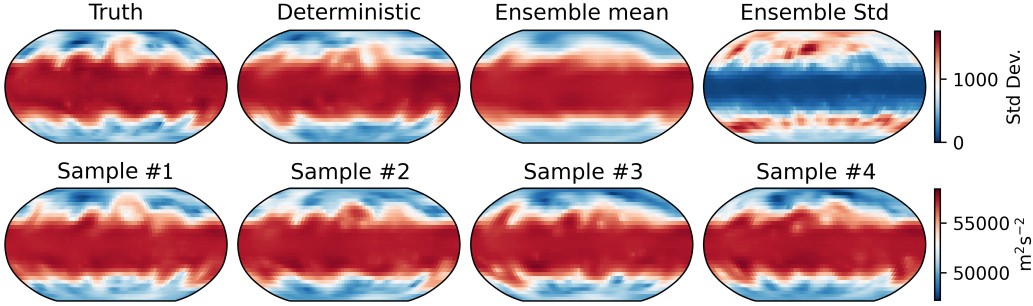

Figure 12: Example forecasts for geopotential at 500 hPa (z500) at lead time 10 days. The forecasts are generated using ARCI-24/6h except for Deterministic which is sampled using the deterministic model. The bottom row shows 4 ensemble members, randomly chosen out of the 50.

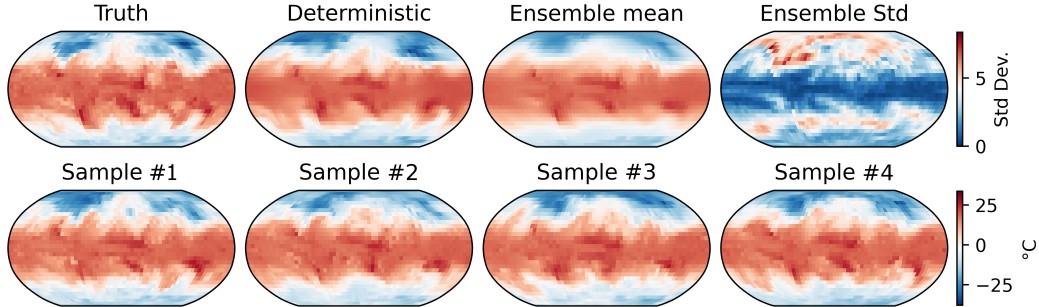

Figure 13: Example forecasts for temperature at 850hPa (t850) at lead time 10 days. The forecasts are generated using ARCI-24/6h except for Deterministic which is sampled using the deterministic model. The bottom row shows 4 ensemble members, randomly chosen out of the 50.

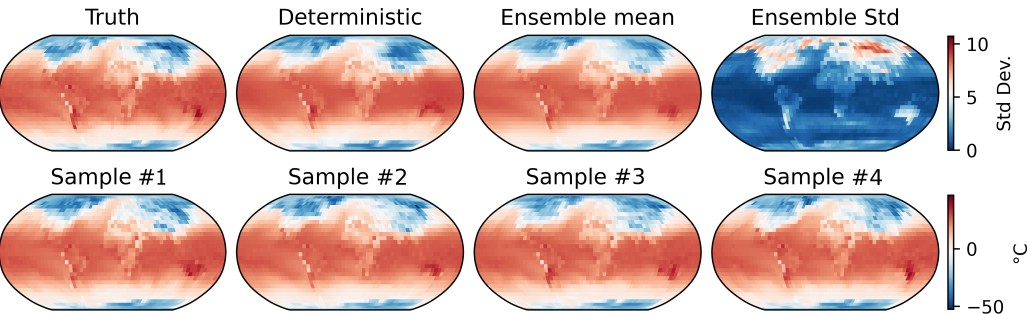

Figure 14: Example forecasts for ground temperature (t2m) at lead time 10 days. The forecasts are generated using ARCI-24/6h except for Deterministic which is sampled using the deterministic model. The bottom row shows 4 ensemble members, randomly chosen out of the 50.

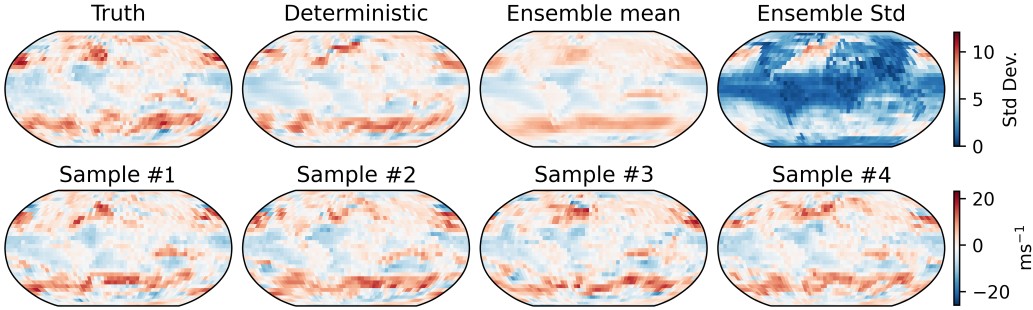

Figure 15: Example forecasts for u-component of wind at 10m (`u10`) at lead time 10 days. The forecasts are generated using ARCI-24/6h except for Deterministic which is sampled using the deterministic model. The bottom row shows 4 ensemble members, randomly chosen out of the 50.

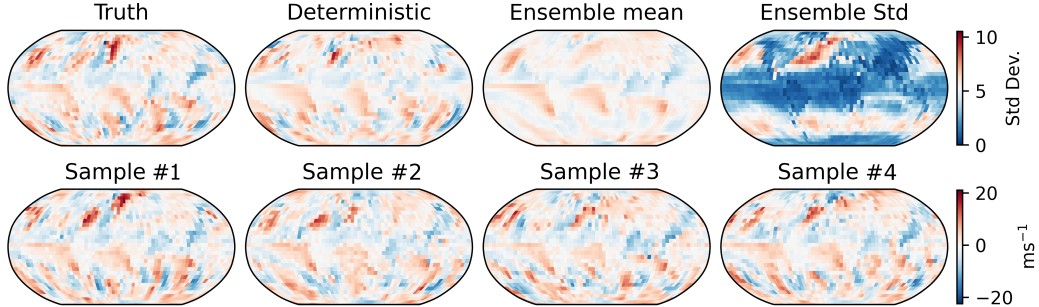

Figure 16: Example forecasts for v-component of wind at 10m (`v10`) at lead time 10 days. The forecasts are generated using ARCI-24/6h except for Deterministic which is sampled using the deterministic model. The bottom row shows 4 ensemble members, randomly chosen out of the 50.

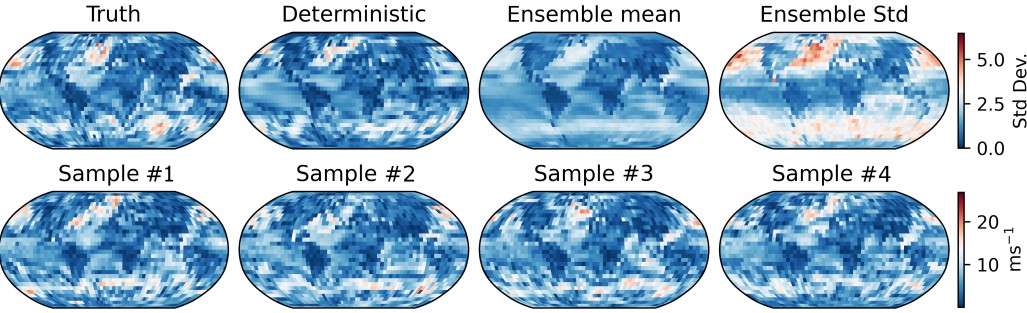

Figure 17: Example forecasts for wind speed at 10m (`ws10`) at lead time 10 days. The forecasts are generated using ARCI-24/6h except for Deterministic which is sampled using the deterministic model. The bottom row shows 4 ensemble members, randomly chosen out of the 50.

# E  CONTINUITY OF SOLUTIONS TO PROBABILITY FLOW ODE

The proposed method in section 4 generates continuous ensemble trajectories by introducing temporal correlations in the noise. Sampling is performed by solving the probability flow ODE:

$$\mathrm{d}z(s) = -sS_\theta(z(s), s; X[k-M:k], t)\mathrm{d}s, \quad z(1) = z_0, \tag{8}$$

starting in pure noise $z_0 \sim p_{\text{noise}}$ and ending in a forecast $z(0) \sim p(X[k+1]|X[k-M:k])$. In practice, the integration is only done up to some time $\epsilon \ll 1$ to overcome instability caused by the singularity of the score function, $\nabla \log p_s$, at $s = 0$.

To ensure that the trajectories are continuous, we must show that solutions to eq. (8) are continuous functions of the lead time $t$. Here, we consider the case where the initial noise $z_0$ remains fixed across different $t$. The more general scenario, where $z_0$ follows a stochastic noise process, is left for future work.

Let $f(s, z, t) := -sS_\theta(z, s; X[k-M:k], t)$ defined for $(s, z) \in D := [\epsilon, 1] \times \mathbb{R}^n$ and $t \in [0, T]$. The dependence on the previous states $X[k-M:k]$ has been suppressed in the notation since all forecasts are conditioned on a fixed $X[k-M:k]$. To phrase the result, we first make an assumption on the regularity of $f$.

**Assumption 1.** *Let $f(s, z, t)$ be continuous in $(s, z, t)$ and locally Lipschitz in $z$, i.e. for any closed bounded set $U \subset D$ there is a $K_U$ such that*

$$|f(s, z_1, t) - f(s, z_2, t)| \le K_U|z_1 - z_2|$$

*for any $(s, z_1), (s, z_2) \in U$ and $t \in [0, T]$.*

If $f(s, z, t)$ is continuous for $(s, z) \in D$, then by the fundamental existence theorem for ODEs (Theorem 1.1 in Hale (2009)), there exists at least one solution of eq. (8) passing through any given point $(s_0, z_0) \in D$. Suppose further this solution is unique and denote it as $z(s, s_0, z_0, t)$. For any $(s_0, z_0) \in D$ and $t \in [0, T]$, let $(a(s_0, z_0, t), b(s_0, z_0, t))$ be the maximal interval of existence of $z(s, s_0, z_0, t)$. We define the set

$$E := \{(s, s_0, z_0, t) : a(s_0, z_0, t) < s < b(s_0, z_0, t), (s_0, z_0) \in D, t \in [0, T]\}.$$

as the *domain of definition* of $z(s, s_0, z_0, t)$. Since our primary interest lies in the continuity of solutions, we assume access to some domain of definition $E$ without explicitly constructing it. For a proof of global existence under stronger assumptions, see theorem B in Section 70, Chapter 13 of Simmons (2016). With this, we can state the main result.

**Theorem 1** (Theorem 3.2 in Hale (2009)). *Assume that $f$ satisfies assumption 1 and consider the initial value problem*

$$\mathrm{d}z(s) = f(s, z, t)\mathrm{d}s, \quad z(s_0) = z_0. \tag{9}$$

*Then, for every $(s_0, z_0) \in D$ and $t \in [0, T]$, there is a unique solution $z(s, s_0, z_0, t)$ with $z(s_0, s_0, z_0, t) = z_0$, and this solution is continuous in $(s, s_0, z_0, t)$ within its domain of definition.*

Theorem 1 shows that the solutions to the probability flow ODE eq. (8) are continuous as functions of lead-time if $-sS_\theta(z, s; X[k-M:k], t)$ is continuous in $(z, s, t)$ and locally Lipschitz in $z$. Since the neural network $S_\theta$ is a composition of continuous functions, it is necessarily a continuous function. The remaining assumption on $S_\theta$ being locally Lipschitz in $z$ is not a particularly strong one, and commonly assumed when proving results about neural networks (Karras et al., 2022; Song et al., 2020; Albergo & Vanden-Eijnden, 2023). To calculate the Lipschitz constant one could for example use the method proposed in Virmaux & Scaman (2018).

