# OpenReview forum: "Continuous Ensemble Weather Forecasting with Diffusion models"
_ICLR.cc/2025/Conference — ICLR 2025 Poster_

### Official Review · Reviewer_PYFz · 2024-10-21

**Soundness:** 2
**Presentation:** 2
**Contribution:** 2
**Rating:** 5
**Confidence:** 2

**Summary:**

This paper introduces Continuous Ensemble Forecasting for global weather forecasting.

**Strengths:**

1. Weather forecasting is an important problem.

2.The method introduced in the paper seems to exhibit a certain level of novelty.

3.Some of the figures in the paper look good.

**Weaknesses:**

1.The paper's presentation seems somewhat disorganized. The introduction does not clearly articulate the limitations of current weather forecasting methods or how the proposed method addresses these limitations. Figure 1, being the first figure, lacks sufficient information; ideally, a single figure should clearly highlight the differences between the proposed method and previous methods, as well as explain how the proposed method overcomes the limitations of prior approaches. In the Methods section, there is a tendency to mix descriptions of existing methods with the new method, which can obscure the exact nature of the contribution made by the proposed method. Figure 2 should also provide a clearer depiction of the methodology.

2.Based on the unclear description provided in the paper, the contributions appear insufficient, and the contributions listed in the introduction seem to be something that any weather forecasting model could achieve.

3.The baseline models compared in the paper are somewhat limited and do not appear to be the latest SOTA models. Moreover, according to Table 1, the ARCI-24/6h model proposed in the paper (if I understand correctly) is not the optimal model under most settings.

**Questions:**

1.Were the limitations of previous methods that they were computationally expensive? How does the method proposed in the paper address the limitations of previous approaches?

2.What specific contributions does the paper make that differentiate it from other weather forecasting models?

---

> ### Author Response · Authors · 2024-11-18
> **Response**
>
> We thank the reviewer for their concerns and have tried to clarify the points raised.
>
> ## Weaknesses:
>
> > The paper's presentation seems somewhat disorganized. The introduction does not clearly articulate the limitations of current weather forecasting methods or how the proposed method addresses these limitations.
>
> In the second paragraph of the introduction we do outline the drawbacks of existing diffusion models for ensemble weather forecasting. Existing models are 1) computationally expensive due to requiring many sequential forward passes through the neural network and 2) have a coarse temporal resolution. As these models are applied autoregressively they tend to accumulate error over time. Increasing the temporal resolution by taking shorter time steps thus results in higher errors. This is remedied by our method (see e.g. figure 6).
>
> > Based on the unclear description provided in the paper, the contributions appear insufficient, and the contributions listed in the introduction seem to be something that any weather forecasting model could achieve.
>
> We respectfully disagree. Most existing weather forecasting models are deterministic and can not produce ensemble forecasts that capture the uncertainty in the dynamics, as we do. There are a few probabilistic models available, as we have discussed in our related work section, but they all differ in key aspects from our method as we explain in more detail in the answer below.
>
> > 2.What specific contributions does the paper make that differentiate it from other weather forecasting models?
>
> Let's revisit the contributions again. We propose a novel method for ensemble weather forecasting built on diffusion that:
> 1. can generate ensemble member trajectories without iteration,
>
> Our model can generate temporally consistent forecast trajectories without ever seeing its own predictions. This had not been considered before our work and thus, there is no existing probabilistic weather forecasting model that can do this. The benefit of this is twofold. Firstly, the parallelization means that computational costs can be distributed on large computing clusters, reducing the time it takes to generate forecasts. Secondly, and more importantly,
> since there is no autoregressive steps, the model does not accumulate errors over time. This property allows us to increase the temporal resolution without sacrificing performance, something SOTA models struggle with.
>
> 2. can forecast arbitrary lead-times,
>
> Almost all existing models act on a fixed temporal resolution. With our framework, you are much more flexible. For example, you might want very high resolution in the beginning of a forecast but a lower resolution after a few days. With purely autoregressive models, you are forced to iterate on the high resolution forecast. With our framework you have the freedom to change resolution across a forecast trajectory without affecting performance of temporal consistency.
>
> 3. can be used together with iteration to improve performance on long rollouts,
>
> Continuous forecasting is effective for forecasting hours to days but can struggle to forecast longer lead times where the correlation is weaker. Autoregressive forecasting excels at long lead times when used with longer (24h) timesteps [1], but comes at the loss of temporal resolution. Our framework generalizes the autoregressive model, allowing both continuous and autoregressive forecasting in a single model. The ARCI method proposed leverages this by iterating on a longer timestep and forecasting the intermediate timesteps using Continuous Ensemble Forecasting.
>
> 4. achieves competitive performance in global weather forecasting.
>
> This is demonstrated in the experiments where the ARCI model outperforms all other models at $\leq$ 6 hour resolution. For a general discussion about baselines, see one of the general comments above.
>
> [1] Bi, Kaifeng, Lingxi Xie, Hengheng Zhang, Xin Chen, Xiaotao Gu, and Qi Tian. “Accurate Medium-Range Global Weather Forecasting with 3D Neural Networks.” Nature 619, no. 7970 (July 2023): 533–38. https://doi.org/10.1038/s41586-023-06185-3.

---

> > ### Author Response · Authors · 2024-11-18
> > **Response Continued**
> >
> > > Figure 1, being the first figure, lacks sufficient information; ideally, a single figure should clearly highlight the differences between the proposed method and previous methods, as well as explain how the proposed method overcomes the limitations of prior approaches.
> >
> > Thank you for this suggestion. We agree that this figure didn't clearly explain the model and have updated it to better increase clarity.
> >
> > > The baseline models compared in the paper are somewhat limited and do not appear to be the latest SOTA models.
> >
> > We have written a general answer answering many of the questions related to baselines which can be found in one of the official comments.
> >
> > > Moreover, according to Table 1, the ARCI-24/6h model proposed in the paper (if I understand correctly) is not the optimal model under most settings.
> >
> > Thank you for bringing up this question as it highlights an important point. Instead of comparing the performance of specific models at different temporal resolutions, we should compare the performance of the AR vs the ARCI class of models. The relevant comparison is thus between ARCI-24/6h and the other models operating at 6 h resolution (all but AR-24h). Within this set of models ARCI achieves substantially lower errors across the board. The AR-24h models is best viewed as a gold-standard reference for what error could be achieved if we do not care about the temporal resolution, take very long autoregressive steps and thus minimize the error accumulation. Thus, the fact that ARCI-24/6h comes close to AR-24h is a key result for our method and what we wanted to show by including AR-24h in the table .
> >
> > Almost all SOTA models use autoregressive forecasting (AR) at different temporal resolutions. This class can obtain very high performance when used at a coarse temporal resolution. For example, the AR-24h model that you pointed out is optimal in table 1 has a temporal resolution of 24h. However, in most cases, you are interested in forecasts with higher temporal resolution. At 6h steps, the AR-6h model performs slightly worse, but still reasonably well. However, as the autoregressive step length becomes even smaller the error accumulation takes over and the model becomes unusable. This can be seen clearly in figure 6 where the performance of AR-1h quickly deteriorates. Because of this, no existing data-driven probabilistic global weather forecasting models produces forecasts at $<$ 6 hour resolution.
> >
> > The ARCI model presented in our paper gets around this by avoiding most of the sequential steps. This means that ARCI can have a temporal resolution up to 1h without sacrificing performance. Thus, the fact that ARCI-24/6h comes even close to AR-24h is a remarkable feat.
> >
> > > In the Methods section, there is a tendency to mix descriptions of existing methods with the new method, which can obscure the exact nature of the contribution made by the proposed method. Figure 2 should also provide a clearer depiction of the methodology.
> >
> > Section 3 covers the problem statement, existing methods (Autoregressive forecasting) and conditional diffusion models. The purpose of this is to provide some necessary context for introducing our method in the next section.
> > Everything in section four should be seen as part of the novel contributions of our paper. This includes Algorithm 1-3 which are all part of the continuous ensemble forecasting framework we propose.
> >
> > However, we recognize that some confusion might arise from section 4.3 where we combine Continuous Ensemble Forecasting with Autoregressive forecasting to propose the ARCI method which combines the best of these methods. We have considered this when updating the paper.
> >
> > Finally, the purpose of figure 2 is not to provide an overview of the methodology but to illustrate the argument in section 4.1. For a clearer picture of the methodology, see the revised figure 1.
> >
> > > 1.Were the limitations of previous methods that they were computationally expensive? How does the method proposed in the paper address the limitations of previous approaches?
> >
> > We have listed the limitations of previous methods in a previous answers where we explain how our model addresses these limitations.
> >
> > We would like to clarify that our model does not achieve a reduction in computational costs in comparison to existing methods, and have updated the paper to clarify this. Instead, what we achieve is parallel sampling of entire forecast trajectories. Using large computing clusters, this can be used to decrease the computational time, but the cost is redistributed rather than reduced.
> >
> > The core advantage of this parallelization instead lies in the reduction of sequential autoregressive steps that accumulate errors over time. As shown in figure 6, this significantly increases the performance of long-range forecasts with high temporal resolution.
> >
> > We hope that our answers addressed all your concerns and that you would consider revising your score.

---

> ### Author Response · Authors · 2024-12-02
> **Reminder**
>
> Dear Reviewer PYFz,
>
> The discussion period ends in a little over 24h. We believe that we have addressed all your concerns. **Please consider replying to our answers.**
>
> Regards,
>
> The authors

---

### Official Review · Reviewer_1nBA · 2024-10-28

**Soundness:** 2
**Presentation:** 2
**Contribution:** 2
**Rating:** 5
**Confidence:** 3

**Summary:**

This work proposes an ODE-based method to address the limitations of previous a single forecasting step and rolled out autoregressively methods for weather prediction. It can generate long-range weather predictions without iteration and forecast arbitrary lead times.

**Strengths:**

- It is interesting to use probability flow ODE of the noise in diffusion models. Leveraging a lead-time-dependent deterministic ODE-solver to generate a smooth and continuous trajectory is meaningful compared to autoregressive methods.
- Authors provide intensive experiments on their method.

**Weaknesses:**

- Need more descriptions on how probability flow ODE works.
- Need more comparisons with other SOTAs.
- No available codes to help readers better understand the paper.

**Questions:**

**Major:**
- It is a good idea to use the probability flow ODE from the work proposed from https://arxiv.org/pdf/2206.00364 in diffusion models. But what is the assumption behind the customized ODE in Eq. (2)? Why is there a linear noise level $sigma$ in Eq. (2)? What if $\sigma^2$? More explanation would help.
- From the paper title and section 4 name, "CONTINUOUS" seems to be used to describe $ensemble forecasting$. However, what the authors really want to convey is the temporal continuity, i.e., any arbitrary fine temporal resolution. You may need to think over the rephrasing. Otherwise, it is confusing for readers (at least for me).
- In Figure 1, do you need to include the 1-hour prediction as the input for the 6-hour prediction? I saw you have explanations in sections 3 and 4. But Figure 1 deserves a more precise description within its title since it is the main framework of your method.
- It would be better to provide more comparisons with previous SOTA ML models. Even though I know not all models did not release source codes to the public, but some did, e.g., FourCastNet and ClimaX.
- The authors mentioned the limitation of diffusion models: computationally expensive. I suggest adding more analysis of computational efficiency.

**Minor:**
- You are using the letter 'k' for different representations (see Eq. 1 and Alg. 1).
- It is customary to denote a time point with the superscript or subscript. It looks like a function if using $X[k+1]$ or $X(\tau)$.
- In the line under Eq. (2), why is $z(1)$ for $X[k+1]$? Isn't it the number of diffusion steps or the noise levels?
- What is $v_i^{k}$ in Algorithm 2?

**Details Of Ethics Concerns:**

Nan

---

> ### Author Response · Authors · 2024-11-18
> **Response**
>
> We thank the reviewer for their insightful questions and do our best of answering them below.
>
> ## Major questions
>
> > It is a good idea to use the probability flow ODE from the work proposed from \href{https://arxiv.org/pdf/2206.00364}{https://arxiv.org/pdf/2206.00364} in diffusion models. But what is the assumption behind the customized ODE in Eq. (2)? Why is there a linear noise level  in Eq. (2)? What if $\sigma^2$? More explanation would help.
>
> The probability flow ODE does not affect the physical time process, just the sampling of each time step. We have updated the paper to reflect that any noise level $\sigma$ can be used during sampling. The choice of a linear $\sigma(s)=s$ is based on Karras et al. [1] due to its favorable sampling properties. A different noise scheduling, such as $\sigma(s)=s^2$ is entirely possible in this framework and would not change anything about the forecasting properties. In fact, we can use any continuous normalizing flow model which maps Gaussian noise to "data samples" by solving an ODE (including probability flow diffusion models, flow matching, stochastic interpolants) in our framework, but chose to use the model by Kerras et al. for simplicity. We have clarified this flexibility in the revised paper.
>
> [1] Karras, Tero, Miika Aittala, Timo Aila, and Samuli Laine. “Elucidating the Design Space of Diffusion-Based Generative Models.” arXiv, October 11, 2022. https://doi.org/10.48550/arXiv.2206.00364.
>
> > From the paper title and section 4 name, "CONTINUOUS" seems to be used to describe Ensemble Forecasting. However, what the authors really want to convey is the temporal continuity, i.e., any arbitrary fine temporal resolution. You may need to think over the rephrasing. Otherwise, it is confusing for readers (at least for me).
>
> We appreciate the concern for confusing terminology. The name continuous forecasting refers to terminology introduced in [2] for learning a forecasting model that can produce forecasts at any time using lead-time as input. This should be contrasted with iterative or direct forecasting where the forecast is iteratively generated or directly predicted with a fixed lead-time model. The name continuous ensemble forecasting refers to the ability to generate ensemble forecasts using continuous forecasting, something never considered using only machine learning before our paper. However, we will spend some time thinking about how domain-specific terminology can be replaced to ease understanding.
>
> [2] Nguyen, Tung, Rohan Shah, Hritik Bansal, Troy Arcomano, Sandeep Madireddy, Romit Maulik, Veerabhadra Kotamarthi, Ian Foster, and Aditya Grover. “Scaling Transformer Neural Networks for Skillful and Reliable Medium-Range Weather Forecasting.” arXiv, December 6, 2023. https://doi.org/10.48550/arXiv.2312.03876.
>
> > In Figure 1, do you need to include the 1-hour prediction as the input for the 6-hour prediction? I saw you have explanations in sections 3 and 4. But Figure 1 deserves a more precise description within its title since it is the main framework of your method.
>
> We thank you for pointing this out and have made immediate efforts in fixing this in the updated paper. Understanding that the 1-hour prediction is not needed as input to the 6-hour prediction is crucial and should be made explicit. We have updated figure 1 and its caption accordingly.
>
> > It would be better to provide more comparisons with previous SOTA ML models. Even though I know not all models did not release source codes to the public, but some did, e.g., FourCastNet and ClimaX.
>
> We have written a general answer answering many of the questions related to baselines which can be found in one of the official comments.
>
> > The authors mentioned the limitation of diffusion models: computationally expensive. I suggest adding more analysis of computational efficiency.
>
> Sampling a 10-day forecast with 6h resolution for a single member from AR-6h takes 32 seconds, but by parallelizing the 6h timesteps in ARCI-24/6h this reduces to 8 seconds. For higher temporal resolution (which is what we focus on) the savings are even larger.
>
> The Heun solver used during sampling uses 40 function evaluations to generate a single forecast. The deterministic baseline would instead generate a forecast in a single evaluation, reducing the 32 seconds to 0.8 seconds for a 10 day forecast. The same argument can be made for a generative latent-variable model, such as Graph-EFM.

---

> > ### Author Response · Authors · 2024-11-18
> > **Response Continued**
> >
> > ## Minor questions
> >
> > >You are using the letter 'k' for different representations (see Eq. 1 and Alg. 1).
> >
> > Thank you for pointing this out, we have changed the superscript in the algorithms to avoid confusion.
> >
> > >It is customary to denote a time point with the superscript or subscript. It looks like a function if using $X[k+1]$ or $X(t)$.
> >
> > In the algorithms, where there are actual realisations $x_k$, we have tried to use lower case letters with subscripts. When referring to the random variable or forecast trajectory, we have opted to use capital letters $X$ for the continuous time $X(t)$ and discrete time $X[k]$ case. While this might cause some confusion at first, this allows us to reason about $X$ as a function that can be evaluated at different times $t$. This is crucial for talking about continuity of trajectories. In particular, $X$ is a stochastic process and the $X(t)$/$X[k]$ notation is customary when discussing these (see e.g. the wikipedia page on discretization, https://en.wikipedia.org/wiki/Discretization).
> >
> > >In the line under Eq. (2), why is $z(1)$ for $X[k+1]$? Isn't it the number of diffusion steps or the noise levels?
> >
> > We've updated the paper to clarify this. In particular, instead of starting at diffusion time $s=0$, we now start at $s=1$ to better match the notation used in other diffusion literature. To generate a sample, we generate some random noise $\boldsymbol{z}(1)$ and iteratively transform it into a sample $\boldsymbol{z}(0)$. This means discretizing the diffusion time interval $s\in[0,1]$ and solving the probability flow ODE for these times. The end result $\boldsymbol{z}(0)$ will then be a sample from $p(X[k{+}1]|X[k{-}M{:}k])$.
> >
> > >What is $v_i^k$ in Algorithm 2?
> >
> > The sample $\nu_i^j$ in algorithm 2 refers to the noise driving the Ornstein-Uhlenbeck process. The purpose of this is to add some extra randomness to the fixed noise $z_i$, as argued in section 4.2. We have added some context for this variable in the paper.
> >
> > > No available codes to help readers better understand the paper.
> >
> > We share the regard for public access code for research papers and will share our code on GitHub after publication.
> >
> > We hope that our answers were satisfactory and that you would consider revising your rating. We would also be happy continuing this discussion in further comments.

---

> > > ### Author Response · Authors · 2024-12-02
> > > **Reminder**
> > >
> > > Dear Reviewer 1nBA,
> > >
> > > The discussion period ends in a little over 24h. We believe that we have addressed all your concerns. **Please consider replying to our answers.**
> > >
> > > Regards,
> > >
> > > The authors

---

### Official Review · Reviewer_c3dx · 2024-10-29

**Soundness:** 2
**Presentation:** 3
**Contribution:** 2
**Rating:** 5
**Confidence:** 3

**Summary:**

The "Continuous Ensemble Weather Forecasting with Diffusion Models" paper introduces an innovative framework for probabilistic weather forecasting using diffusion models. It focuses on enhancing computational efficiency by removing the autoregressive nature of sampling from diffusion models.

**Strengths:**

The model developed in this paper presents an approach that eliminates traditional autoregressive steps during sampling from diffusion models, allowing it to achieve high temporal resolution and efficiency in forecasting. This is an exciting approach for improving computational efficiency, a feature that has been challenging to achieve in weather prediction.

The paper's application is exciting, as diffusion models are typically used in areas like image generation rather than meteorological forecasting.

**Weaknesses:**

The paper lacks references to closely related methods, such as Zheng et al. (2023) on fast sampling of diffusion models via operator learning, which shares similarities with the proposed model’s sampling approach. It would be beneficial to clarify how the proposed sampling method diverges from these approaches.

Although not directly used for weather data, recent models like Dyfusion, Alternator, and Mamba (with s-mamba used specifically for weather forecasting) have shown promise in similar domains, such as sea surface temperature forecasting. Including these models as baselines or discussing their relevance could offer a more comprehensive comparison, illustrating where this model stands relative to other contemporary approaches.

The paper mentions “regularity conditions” (e.g., Line 234) without specifying them rigorously. These conditions are fundamental to claims about trajectory continuity and temporal stability of forecasts. A formal mathematical description of these conditions, alongside proof or empirical validation, would be necessary to substantiate these claims.

The claims regarding fixed noise across time steps and ODE-based interpolation are under-supported. No mathematical evidence or ablation study is presented to show the impact of fixing noise on forecasting performance. Including such a study could clarify the advantages of this design choice.

The proposed Algorithm 2 seems tailored to a specific diffusion model setup. It would be valuable to discuss whether this algorithm generalizes to other diffusion models or is fundamentally limited to the framework outlined in the paper. A broader discussion on this topic could enhance the paper’s applicability to related models.

The results lack completeness, as only two of the five variables used for forecasting are presented in Table 2. Including comparisons for all five variables would provide a clearer picture of the model's strengths and weaknesses across different atmospheric variables.

The absence of error bars due to computational limitations weakens the robustness of the reported results. Given the probabilistic nature of the model, showcasing variability over multiple runs would offer greater confidence in the findings. Including error bars or confidence intervals for at least a subset of key results would be helpful, even if computational constraints prevent a complete assessment.

Numerous claims about model performance, such as trajectory continuity, lack rigorous mathematical proof and appear to rely on empirical observations. Addressing this would lend greater theoretical robustness to the proposed model.

There is no mathematical derivation for the training objective and loss function. A step-by-step derivation of these, with explanations of any modifications relative to standard diffusion model objectives and the rationale for these modifications in the weather forecasting context, would clarify the theoretical foundation of the approach.

**Questions:**

Could you clarify the rationale behind using fixed noise across time steps? Additionally, has any sensitivity analysis or ablation study been conducted to validate this design choice? If so, please provide these findings.

Could you further explain how interpolation using an ODE solver contributes to the model’s performance, and how it compares to alternative interpolation methods?

Only two of the five forecasting variables are included in Table 2. Could you explain the rationale for this selective presentation and consider including results for all five variables to enable a comprehensive evaluation?

Could you provide formal proofs or derivations for key claims, particularly those regarding trajectory continuity and temporal stability?

---

> ### Author Response · Authors · 2024-11-18
> **Response**
>
> We thank the reviewer for their comments and have addressed all of them below.
>
> # Weaknesses
>
> ## Relation to Zheng et al.
> > The paper lacks references to closely related methods, such as Zheng et al. [1] on fast sampling of diffusion models via operator learning, which shares similarities with the proposed model’s sampling approach. It would be beneficial to clarify how the proposed sampling method diverges from these approaches.
>
> Thank you for bringing this work to our attention. While Zheng et al. considers a different problem, our work share some interesting connections. They consider a neural operator approach to speed up sampling from diffusion models. This allows them to generate the entire diffusion trajectory in parallel, from a single noise sample.
>
> There are some key differences to our work. Zheng et al. considers sampling of static images and do not have a physical time that drives the system forward. Their parallel sampling is done on the diffusion time axis, while we act on the physical time axis. We still solve the probability flow ODE at each time-step, and make no attempt to parallelize this process. This is perhaps best clarified by our updated version of Figure 1, where the work of Zheng et al. is concerned with speeding up the vertical probability flow solver. Our work still uses the method from Karras et al. for this vertical diffusion sampling, but parallelize generation over the horizontal time axis.
>
> Note that we could use the approach by Zheng et al. in our method as an alterative to Karras et al. to reduce the computational cost of the probability flow sampling. Similar things can be noted about changing the diffusion to other generative models that use the probability flow ODE, such as flow matching or stochastic interpolants.
>
> Since the key contributions of our work are independent of the generative method used, as long as it solves the probability flow ODE, we chose the method by Karras et al. for brevity. We have clarified the flexibility in this design choice in the paper.
>
> [1] Zheng, Hongkai, Weili Nie, Arash Vahdat, Kamyar Azizzadenesheli, and Anima Anandkumar. “Fast Sampling of Diffusion Models via Operator Learning.” In Proceedings of the 40th International Conference on Machine Learning, 42390–402. PMLR, 2023. https://proceedings.mlr.press/v202/zheng23d.html.

---

> > ### Author Response · Authors · 2024-11-18
> > **Response Continued**
> >
> > ## Baselines
> >
> > > Although not directly used for weather data, recent models like Dyfusion, Alternator, and Mamba (with s-mamba used specifically for weather forecasting) have shown promise in similar domains, such as sea surface temperature forecasting. Including these models as baselines or discussing their relevance could offer a more comprehensive comparison, illustrating where this model stands relative to other contemporary approaches.
> >
> > DYffusion [2] is indeed a related model that we discuss in some detail in the related work section. We appreciate your suggestion and will do our best to implement DYffusion in the weather forecasting setup to enable a numerical comparison.
> >
> > The S-mamba model introduced in [3] is a general time-series model and is not adapted to spatio-temporal data. They do apply their method to a weather dataset, but this is using 21 variables at one observation station and is very different from the global spatio-temporal forecasting that we consider (see also our reply to reviewer koDm).  The Alternator, introduced very recently in [4], is a latent-variable model that have shown promising results for sea-surface temperature forecasting. We believe latent-variable models are an interesting class of models since they have computational benefits compared to diffusion models. Because of this, we have chosen to compare to Graph-EFM [5], a SOTA latent-variable model for probabilistic weather forecasting. This model is chosen because it's 1) specifically developed for weather, 2) set to appear as a [spotlight at NeurIPS](https://nips.cc/virtual/2024/poster/93149) and crucially 3) have publicly available code. To the best of our knowledge, the latter point is not true for the Alternator which made a numerical comparison difficult, considering  that this model appeared on arXiv as late as May 2024.
> >
> > We have also written a general answer answering many of the questions related to baselines which can be found in one of the official comments.
> >
> > [2] Cachay, Salva Rühling, Bo Zhao, Hailey James, and Rose Yu. “DYffusion: A Dynamics-Informed Diffusion Model for Spatiotemporal Forecasting,” 2023. https://openreview.net/forum?id=WRGldGm5Hz&noteId=jDWiuqrB98.
> >
> > [3] Wang, Zihan, Fanheng Kong, Shi Feng, Ming Wang, Xiaocui Yang, Han Zhao, Daling Wang, and Yifei Zhang. “Is Mamba Effective for Time Series Forecasting?” arXiv, April 27, 2024. https://doi.org/10.48550/arXiv.2403.11144.
> >
> > [4] Rezaei, Mohammad Reza, and Adji Bousso Dieng. “Alternators For Sequence Modeling.” arXiv, May 20, 2024. https://doi.org/10.48550/arXiv.2405.11848.
> >
> > [5] Oskarsson, Joel, Tomas Landelius, Marc Peter Deisenroth, and Fredrik Lindsten. “Probabilistic Weather Forecasting with Hierarchical Graph Neural Networks.” arXiv, October 26, 2024. https://doi.org/10.48550/arXiv.2406.04759.

---

> > > ### Author Response · Authors · 2024-11-18
> > > **Response Continued**
> > >
> > > ## Proof of trajectory continuity
> > >
> > > > The paper mentions “regularity conditions” (e.g., Line 234) without specifying them rigorously. These conditions are fundamental to claims about trajectory continuity and temporal stability of forecasts. A formal mathematical description of these conditions, alongside proof or empirical validation, would be necessary to substantiate these claims.
> > >
> > > > The claims regarding fixed noise across time steps and ODE-based interpolation are under-supported. No mathematical evidence or ablation study is presented to show the impact of fixing noise on forecasting performance. Including such a study could clarify the advantages of this design choice.
> > >
> > > > Numerous claims about model performance, such as trajectory continuity, lack rigorous mathematical proof and appear to rely on empirical observations. Addressing this would lend greater theoretical robustness to the proposed model.
> > >
> > > > Could you provide formal proofs or derivations for key claims, particularly those regarding trajectory continuity and temporal stability?
> > >
> > > The reason for why we only loosely mentioned "regularity conditions" is that we only made loose claims about continuity (i.e. this was not stated as a theorem or similar). However, we thank you for pointing out this shortcoming of the paper and fully agree that the paper will be much stronger with a formal proof of continuity. We have updated the paper with a proof of this property. See one of the official comments above for more comments about this.
> > >
> > > The reason there is no ablation study on the impact of fixed noise on forecasting performance is because there is no impact. Fixing the noise doesn't affect the marginal distributions at different lead times, just the temporal dependencies in the trajectories. Thus the results in table 1 and figure 3, 6 are valid for both fixed and non-fixed noise. This is an important point and we have updated our paper to clarify this further.
> > >
> > > Regarding empirical validation for trajectory continuity: this is provided in figure 5. When using uncorrelated noise at each step, the temporal difference $\Delta X = |X_{t+1}-X_t|$ quickly diverges. When using fixed or correlated noise, this difference stays fairly constant over time, empirically showing the trajectory continuity.
> > >
> > > Regarding temporal stability, it should be noted that the core algorithms for continuous ensemble forecasting (Alg 1 and Alg 2) make direct forecasts at all leadtimes. Thus, there is no iteration over leadtimes that could result in temporal instability. When we later combine this with auto-regressive rollouts (Alg 3), we do apply the method iteratively and there is a risk that there is an "unstable" accumulation of errors. However, this is no different than any other autoregressive forecasting or diffusion model, such as GenCast [6], and to the best of our knowledge none of the state-of-the-art methods in this application domain (GenCast, GraphCast, ...) provide any formal proofs of temporal stability, and we believe that such a proof is beyond the scope of the current work.
> > >
> > > We would also like to point out that autoregressive stability is likely less of an issue for weather forecasting than for an application like video generation. The best performing method in our evaluation is using 24h steps in the autoregression, which means that there are only 10 iterations for a 10 day forecast. This is roughly the maximum lead time where we can expect skillfull forecasts due to the chaotic nature of the weather. In fact, one of the main advantages of the proposed method (ARCI) is that it enables high temporal resolution, while at the same time using relatively few autoregressive steps to reduce the accumulation of error (see e.g. comparison between ARCI-24/6h and AR-6h in Table 1).

---

> > > > ### Author Response · Authors · 2024-11-18
> > > > **Response Continued**
> > > >
> > > > ## General concerns
> > > >
> > > > > The proposed Algorithm 2 seems tailored to a specific diffusion model setup. It would be valuable to discuss whether this algorithm generalizes to other diffusion models or is fundamentally limited to the framework outlined in the paper. A broader discussion on this topic could enhance the paper’s applicability to related models.
> > > >
> > > > We agree that discussing the applicability of the framework to other diffusion models would be valuable and thank the reviewer for this suggestion. Algorithm 2 is actually quite independent of the diffusion setup. Note that **line 5 in Algorithm 2 does not refer to the noising process in a diffusion model**. Instead this is the stochastic process that we use to correlate the noise as an extension of the fixed noise model. The diffusion sampling is handled by the ODE-solver in line 7. Thus, our framework straightforwardly generalizes to any continuous normalizing flow model that maps Gaussian noise to data  (including probability-flow diffusion, flow matching, and stochastic interpolants), simply by using the corresponding ODE-solver in line 7 . We have added a discussion about this to the paper.
> > > >
> > > > > The results lack completeness, as only two of the five variables used for forecasting are presented in Table 2. Including comparisons for all five variables would provide a clearer picture of the model's strengths and weaknesses across different atmospheric variables.
> > > >
> > > > > Only two of the five forecasting variables are included in Table 2. Could you explain the rationale for this selective presentation and consider including results for all five variables to enable a comprehensive evaluation?
> > > >
> > > > The results for all variables are given in table 6 in appendix C. The z500 and t850 fields displayed in the main article are often used as reference fields when comparing models since they offer valuable insights into atmospheric conditions (e.g. see https://charts.ecmwf.int/products/medium-z500-t850). The 500 hPa level, situated in the mid-troposphere, reveals the large-scale flow patterns that steer weather systems. By examining the height contours, one can identify troughs, ridges, and areas of strong winds. The 850 hPa level, closer to the Earth's surface, provides information about temperature and moisture distribution. This level is particularly useful for distinguishing between warm and cold air masses and locating frontal zones, which are key factors influencing local weather conditions.
> > > >
> > > > > The absence of error bars due to computational limitations weakens the robustness of the reported results. Given the probabilistic nature of the model, showcasing variability over multiple runs would offer greater confidence in the findings. Including error bars or confidence intervals for at least a subset of key results would be helpful, even if computational constraints prevent a complete assessment.
> > > >
> > > > We share the reviewers regard for error estimates and will do our best to provide error bars from multiple re-trained models during the rebuttal period.
> > > >
> > > > It should also be mentioned that the probabilistic properties of the model are readily evaluated using metrics such as CRPS and SSR. This does not tell you anything about the error caused by the randomness in the training, but answers whether the forecast has a realistic error growth over time.
> > > >
> > > > > There is no mathematical derivation for the training objective and loss function. A step-by-step derivation of these, with explanations of any modifications relative to standard diffusion model objectives and the rationale for these modifications in the weather forecasting context, would clarify the theoretical foundation of the approach.
> > > >
> > > > The diffusion model used is a standard conditional diffusion model based on the one introduced in Karras et al. [6] with the same training objective and loss function. In fact, the code is based on their implementation of a diffusion model, and all changes to the framework are listed in Appendix B. We refer to [6] for an excellent derivation of the training objective and loss function.
> > > >
> > > > [6] Karras, Tero, Miika Aittala, Timo Aila, and Samuli Laine. “Elucidating the Design Space of Diffusion-Based Generative Models.” arXiv, October 11, 2022. https://doi.org/10.48550/arXiv.2206.00364.

---

> > > > > ### Author Response · Authors · 2024-11-18
> > > > > **Response Continued**
> > > > >
> > > > > # Questions
> > > > > > Could you clarify the rationale behind using fixed noise across time steps? Additionally, has any sensitivity analysis or ablation study been conducted to validate this design choice? If so, please provide these findings.
> > > > >
> > > > > The rationale behind using fixed noise across time steps is given in the second paragraph in section 4 and further motivated in section 4.1. This builds on a formulation of stochastic dynamics as a distribution over possible forecasting models. By fixing the noise, you essentially pick one of the possible forecasting models, and can use it to forecast an entire trajectory. If the parameterized score function $S_\theta$ has learned to match the theoretical score function $\nabla\log p$, the sampled forecasts should follow the same distribution as the theoretical ones. Thus, the sampled trajectories should reflect actual ensemble trajectories. This builds on the proof of trajectory continuity which has been discussed in a previous answer.
> > > > >
> > > > > Verification of this design choice is provided in figure 5. When using independent noise at each step, the temporal difference $\Delta X = |X_{t+1}-X_t|$ quickly diverges implying that the forecasts are uncorrelated.
> > > > > When using fixed or correlated noise, this difference stays fairly constant over time, empirically showing the trajectory continuity.
> > > > >
> > > > > > Could you further explain how interpolation using an ODE solver contributes to the model’s performance, and how it compares to alternative interpolation methods?
> > > > >
> > > > > The purpose of the ODE-solver is to solve the probability flow ODE that transforms noise into forecasts. The probability flow ODE is important because it gives access to a deterministic continuous mapping from noise to a one-step forecast $X[k+1]$, which is crucial in getting trajectory continuity. However, this solver acts on the vertical diffusion time axis in figure 1, and is thus not related to interpolation on the horizontal time axis. There is no ODE in the physical time domain. We have updated figure 1 to better reflect this.
> > > > >
> > > > > For a discussion on interpolation methods as an alternative to continuous ensemble forecasting, you can look at figure 9 where ARCI is compared with a linear interpolation between 6 hour forecasts. However, we emphasize again that this is unrelated to the ODE-solver.
> > > > >
> > > > > We hope that our answers have addressed all your concerns and that you will consider revising your score. We are also very happy to continue the discussion further.

---

> > > > > > ### Author Response · Authors · 2024-12-02
> > > > > > **Reminder**
> > > > > >
> > > > > > Dear Reviewer c3dx,
> > > > > >
> > > > > > The discussion period ends in a little over 24h. We believe that we have addressed all your concerns. **Please consider replying to our answers.**
> > > > > >
> > > > > > Regards,
> > > > > >
> > > > > > The authors

---

> ### Comment · Reviewer_c3dx · 2024-12-02
>
> Thank you to the authors for their response. While it addresses some of my concerns, I have decided to raise my score to 5.

---

> > ### Author Response · Authors · 2024-12-02
> > **Response**
> >
> > Dear reviewer c3dx,
> >
> > Thank you for your answer and for raising your score! Is it possible to elaborate which of your concerns remain?
> >
> > Thanks again,
> >
> > The authors

---

> > > ### Author Response · Authors · 2024-12-03
> > > **Implementation of DYffusion baseline**
> > >
> > > Dear reviewer c3dx,
> > >
> > > We have added a general comment on the implementation of the DYffusion baseline as promised in the answers to you. I hope this helps in resolving any remaining concerns.
> > >
> > > Thanks again,
> > >
> > > The authors

---

### Official Review · Reviewer_koDm · 2024-11-03

**Soundness:** 2
**Presentation:** 3
**Contribution:** 2
**Rating:** 5
**Confidence:** 2

**Summary:**

This paper uses the diffusion model for continuous ensemble weather forecasting without requiring autoregressive steps to produce future values. As introduced in Section 4.2, the main idea is to replace the fixed Z with a stochastic process Z(T) with continuous sample trajectories.

**Strengths:**

The proposed method is presented clearly.

**Weaknesses:**

W1: there have been exsiting works for nonautoregressive sequence forecasting by diffusion models. However, realted investigation and discussions are missing, e.g.,

[1] NeurIPS'21 CSDI Conditional Score-based Diffusion Models for Probabilistic Time Series Imputation

[2] ICML'23 Non-autoregressive Conditional Diffusion Models for Time Series Prediction

W2: moreover, some questions and details are not provided.
- it would be better to explain what ensemble forecasting is formally
- it is still not very convincing why a stochastic process is better than random Gaussian noises for nonautoregressive generation. Because we can simply represent the forecast outputs as a vector of a length N. Then we can use a Gaussian vector to yield forecasts.
- is there a significant difference between weather forecasting and general time series forecasting?
- what is the input size (the number of variables) of the model in the experiments? It seems there are many variables in the weather forecasting task. Did you perform univariate forecasting tasks for each variable?
- how to fuse $n_{ens}$ trajectories to get the final predictions?

**Questions:**

In the weakness section.

---

> ### Author Response · Authors · 2024-11-18
> **Response**
>
> We thank the reviewer for their comments and do our best to address questions and weaknesses.
>
> > As introduced in Section 4.2, the main idea is to replace the fixed Z with a stochastic process Z(T) with continuous sample trajectories.
>
> Note that replacing $Z$ with $Z(T)$ in section 4.2 is a generalization of the method in section 4.1. The core contribution is instead to use correlated (fixed or autocorrelated) noise instead of uncorrelated noise. This is what allows the trajectories to be continuous.
>
> # Weaknesses
> > W1: there have been exsiting works for nonautoregressive sequence forecasting by diffusion models. However, realted investigation and discussions are missing, e.g.,
>
> >[1] NeurIPS'21 CSDI Conditional Score-based Diffusion Models for Probabilistic Time Series Imputation
>
> The reference refers to imputation rather than the prediction task considered in our work.
>
> ## Weather forecasting vs General time series forecasting
> > Is there a significant difference between weather forecasting and general time series forecasting?
>
> > What is the input size (the number of variables) of the model in the experiments? It seems there are many variables in the weather forecasting task.
>
> Conceptually, weather forecasting is a type of time series prediction, but the fact that we explicitly represent the weather state at multiple spatial locations (with strong spatial dependencies) makes the problem very high dimensional. The ERA5 data we use in our numerical evaluation is using 5 variables for each grid cell in a 32 times 64 grid, resulting in 10 240 variables at each timestep. This is in stark contrast with the weather dataset considered in [2], which contains measurements of 21 variables at one observation station.
>
> Generating 10 minute forecasts up to 1 week in a single step, as done in [2], thus requires predicting 14 112 variables. Applying the method proposed in [2], of predicting the entire time-series at once, in our setting, would require a massive 2 457 600 variables (for 1h forecasts up to 10 days).  While this is perhaps manageable, it should be noted that we have studied a very coarse representation of the atmosphere with just a few selected variables for simplicity. In an operational weather forecasting implementation, we would need to use a much higher spatial resolution and many more spatial fields, which means predicting several steps at once would limit scalability too much for it to be useful in practice. As an example of this, the SOTA probabilistic machine learning weather forecasting model [3] uses a high-resolution version of ERA5 in a 1440 times 720 grid with 84 variables per grid point. Essentially all state-of-the-art MLWP models are based on a 2nd order autoregression for the temporal dimension for this reason.
>
> Nevertheless, we thank you for pointing out the lack of references to the general time-series forecasting literature, where the reference [2] is indeed an excellent method for probabilistic time-series prediction. We will update the related work section to more clearly relate our work to the general time-series forecasting when revising the paper.
>
> [2] ICML'23 Non-autoregressive Conditional Diffusion Models for Time Series Prediction
>
> [3] Price, Ilan, Alvaro Sanchez-Gonzalez, Ferran Alet, Tom R. Andersson, Andrew El-Kadi, Dominic Masters, Timo Ewalds, et al. “GenCast: Diffusion-Based Ensemble Forecasting for Medium-Range Weather.” arXiv, May 1, 2024. https://doi.org/10.48550/arXiv.2312.15796.
>
> > Did you perform univariate forecasting tasks for each variable?
>
> As pointed out above, the variables in our dataset are essentially "*pixel channels*" in a spatial grid. Since the data is highly spatially correlated, predicting each point on its own is not a viable option since it would loose all spatial consistency and the spatial information that is needed for making accurate predictions. In fact, it would be similar to attempting to do video prediction based on a single channel at a single pixel individually. There are also strong correlations across the different variables (channels) in the data, and we forecast all of them jointly with one single model.
>
> >  It is still not very convincing why a stochastic process is better than random Gaussian noises for nonautoregressive generation. Because we can simply represent the forecast outputs as a vector of a length N. Then we can use a Gaussian vector to yield forecasts.
>
> You are absolutely correct that a stochastic process doesn't make sense in the context of [2]. However, when we are restricted to predicting just a single forecast at a time, the stochastic noise process that drives the diffusion sampling process at different lead times is what allows the model to generate temporally consistent trajectories, despite never seeing its own predictions. If we would independently sample Gaussian noise for each lead time we would end up with a set of reasonable predictions, but that do not make sense together as a weather state trajectory.

---

> > ### Author Response · Authors · 2024-11-18
> > **Response Continued**
> >
> > ## Ensemble Forecasting
> > > It would be better to explain what ensemble forecasting is formally
> >
> > Thank you for pointing out this. In the context of weather forecasting, ensemble forecasting refers to generating multiple possible weather trajectories. This set (or ensemble) of forecasts helps show the range of possible future weather conditions. In this paper, we define it as samples from the probability distribution of future states given information about previous weather states, as outlined in the problem formulation in section 3. It is a well established terminology in the weather forecasting literature, but we will make sure to clarify this to make the paper more accessible to the broader machine learning community.
> >
> > > How to fuse  trajectories to get the final predictions?
> >
> > The question of combining trajectories into a single one is an interesting one. A simple way to combine the forecasts is to take the mean of the ensemble. This is the forecast used for the RMSE scores presented in the paper. However, since the distribution can be highly non-Gaussian with intricate spatial dependencies and possibly multimodal, the ensemble mean is blurry and unrealistic as seen in Fig. 4.
> >
> > Moreover, for meteorologists, having access to an ensemble of forecasts means being able to assess multiple possible futures. One can imagine this being important for risk mitigation and decision making, for example when predicting the path of a tropical cyclone. In general, the "fusion" should thus ideally be done at a later stage, for instance by using the entire ensemble to estimate the probability of a cyclone hitting land (as an example).
> >
> > We hope that your concerns have been adequately addressed, and if so, you would consider revising your score. We would also be happy to discuss further.

---

> > > ### Author Response · Authors · 2024-12-02
> > > **Reminder**
> > >
> > > Dear Reviewer koDm,
> > >
> > > The discussion period ends in a little over 24h. We believe that we have addressed all your concerns. **Please consider replying to our answers.**
> > >
> > > Regards,
> > >
> > > The authors

---

### Author Response · Authors · 2024-11-18
**General response about proof of trajectory continuity**

In this paper we have presented a novel method for generating forecast trajectories in a nonsequential fashion, leveraging a stochastic process in the Gaussian noise. Empirically, we show in figure 5 that this generates temporally consistent and continuous forecast trajectories. This can also be seen in the supplementary animations.

However, as pointed out by the reviewers, this claim lacks substantial theoretical proof. In section 4.1 we provide some intuition for why the trajectories would be continuous with enough data and training. This analysis relies heavily on the assumption that the solution to eq. 2 preserves this continuity. In the paper, we claim that this will hold under some regularity conditions but leave this for future work. As this was highlighted by the reviewers as something important, we have spent some time investigating this and provided a proof of this property. **The specific assumptions and proof of desired properties is given in appendix E of the revised paper**.

---

### Author Response · Authors · 2024-11-18
**General response about baselines**

We'd like to stress the **importance of using ensemble forecasting models as baselines**. Many of the proposed baselines (FourCastNet [1], ClimaX [2] etc) are deterministic models that can not generate ensemble forecasts without perturbing initial conditions¹. We have included a deterministic baseline trained using the same backbone U-Net as the diffusion model, which we outperform significantly. In fact, the contribution of our paper is not to provide a new backbone architecture, but rather the development of a diffusion-model-based framework for enabling time-continuous ensemble predictions. We can in principle use another backbone architecture in our framework, including GNN-based (GraphCast [3]), Transformer-based (ClimaX), or Neural-operator-based (FourCastNet),  but focused on a U-Net for simplicity.

We believe latent-variable models are an interesting class of models since they have computational benefits compared to diffusion models. Because of this, we have chosen to compare to Graph-EFM [4], a SOTA latent-variable model for probabilistic weather forecasting. This model is chosen because it's 1) specifically developed for weather, 2) set to appear as a [spotlight at NeurIPS](https://nips.cc/virtual/2024/poster/93149) and crucially 3) have publicly available code. Their model outperforms the GraphCast model which in turn has shown to be superior to other models such as Fourcastnet and ClimaX. Thus we believe that outperforming Graph-EFM is a great achievement.

We would also like to highlight the connection between our model and the SOTA weather forecasting model GenCast [5]. The AR models used as baselines in the paper is the exact forecasting setup from [5]. The difference is that it's implemented with a U-net backbone instead of a graph neural network. The reason for this is that we want to use the same backbone architecture for the different diffusion model variants that we consider in the paper for a fair comparison, and to avoid confounding the effect of the backbone architecture with the effect of different diffusion model setups. In the paper we show that the ARCI model outperforms the GenCast setup for high temporal resolution, making this a SOTA model for global weather forecasting at 1h resolution. We have updated the paper to clarify this connection between the AR model and GenCast.

Howevever, as suggested by reviewer c3dX, there are other similar models that haven't been applied to weather forecasting. DYffusion [6] is a related model that we discuss in some detail in the related work section. We will try to implement DYffusion in the weather forecasting setup and hope to share the results with you during the discussion period.

[1] Kurth, Thorsten, Shashank Subramanian, Peter Harrington, Jaideep Pathak, Morteza Mardani, David Hall, Andrea Miele, Karthik Kashinath, and Anima Anandkumar. “FourCastNet: Accelerating Global High-Resolution Weather Forecasting Using Adaptive Fourier Neural Operators.” In Proceedings of the Platform for Advanced Scientific Computing Conference, 1–11. PASC ’23. New York, NY, USA: Association for Computing Machinery, 2023. https://doi.org/10.1145/3592979.3593412.

[2] Nguyen, Tung, Johannes Brandstetter, Ashish Kapoor, Jayesh K. Gupta, and Aditya Grover. “ClimaX: A Foundation Model for Weather and Climate.” In Proceedings of the 40th International Conference on Machine Learning, 25904–38. PMLR, 2023. https://proceedings.mlr.press/v202/nguyen23a.html.

[3] Lam, Remi, Alvaro Sanchez-Gonzalez, Matthew Willson, Peter Wirnsberger, Meire Fortunato, Ferran Alet, Suman Ravuri, et al. “Learning Skillful Medium-Range Global Weather Forecasting.” Science 382, no. 6677 (December 22, 2023): 1416–21. https://doi.org/10.1126/science.adi2336.

[4] Oskarsson, Joel, Tomas Landelius, Marc Peter Deisenroth, and Fredrik Lindsten. “Probabilistic Weather Forecasting with Hierarchical Graph Neural Networks.” arXiv, October 26, 2024. https://doi.org/10.48550/arXiv.2406.04759.

[5] Price, Ilan, Alvaro Sanchez-Gonzalez, Ferran Alet, Tom R. Andersson, Andrew El-Kadi, Dominic Masters, Timo Ewalds, et al. “GenCast: Diffusion-Based Ensemble Forecasting for Medium-Range Weather.” arXiv, May 1, 2024. https://doi.org/10.48550/arXiv.2312.15796.

[6] Cachay, Salva Rühling, Bo Zhao, Hailey James, and Rose Yu. “DYffusion: A Dynamics-Informed Diffusion Model for Spatiotemporal Forecasting,” 2023. https://openreview.net/forum?id=WRGldGm5Hz&noteId=jDWiuqrB98.

¹ They can produce a type of ensemble forecasts by perturbing initial conditions, but this captures a different type of uncertainty than what we are focusing on, namely the uncertainty in the dynamics/forecasts even for fixed initial conditions. Note that our method can also be used with initial condition perturbation for capturing both types of uncertainties, but we have not explored this combination to avoid confounding the results.

---

### Author Response · Authors · 2024-11-25
**Gentle Reminder**

Dear Reviewers, Area Chair,
The discussion period ends in two days. We believe that we have addressed all your concerns. See the updated draft, our general reply, and the specific answers we sent you. We would love to hear your thoughts on these updates and provide further clarifications if needed.

---

### Author Response · Authors · 2024-11-28
**Message to reviewers**

Dear reviewers,

We have updated the draft with **highlights of all the changes we made to the original draft**, making it easier for you to find the relevant changes. This includes a calculation of error bars from multiple re-trained models in appendix C, as suggested by reviewer c3dx. We have also added a more detailed discussion on the limitations of the DYffusion model which was mentioned by reviewer c3dx as a related model. We are still working on a full retraining and evaluation of DYffusion in the weather forecasting setup and hope to share the results with you before the discussion period ends.

We believe that we have addressed all your concerns. See the updated draft, our general replies, and the specific answers we sent you. We would love to hear your thoughts on these updates and provide further clarifications if needed.

Regards,

The authors

---

### Author Response · Authors · 2024-12-03
**Implementation of DYffusion baseline**

Dear Reviewers, Area Chair,

Almost all current probabilistic spatio-temporal models rely on autoregressive rollouts that struggle at high temporal resolution. This includes the current SOTA GenCast model [1]. In this paper we have presented a novel method that scales well to arbitrarily fine temporal resolutions. Another alternative to autoregressive rollouts is the DYffusion framework [2], where stochastic interpolation and deterministic forecasting is combined into a diffusion-like model. The method allows for forecasting at arbitrary temporal resolution, but still requires sequential computations for sampling the prediction. **As promised in the answers to reviewer c3dx**, we have worked on a full retraining and evaluation of DYffusion in the weather forecasting setup. This is the result of these efforts.

**We have adapted the DYffusion codebase to work with weather forecasting using ERA5 data.** This application was not considered in the original DYffusion papers which focused on sea-surface forecasting [2] and climate modeling [3]. To ensure a fair evaluation, we have used the same backbone U-Net as our diffusion model, and a lat-lon weighted mean-squared-error loss. We have trained it with a horizon of 24h, with 1h time-steps, making it directly comparable to ARCI-24/1h. For the hyperparameters, we were not able to do an extensive search due to the time constraints, and instead used values similar to those proposed in the original paper. Below we evaluate DYffusion on the 2018 test data and compare it to the ARCI-24/1h model.

| | | | 5 days | | | 10 days | |
| -- | --  | -- | -- | -- | -- | -- | -- |
| Variable | Model      | RMSE  | CRPS  | SSR  | RMSE   | CRPS  | SSR  |
| z500  | DYffusion     | 633.5 | 303.7 | 0.93 | 823.1  | 393.6 | 1.00 |
| | ARCI-24/1h    | 564   | 264.4 | 0.88 | 776.1  | 369   | 0.95 |
| t850  | DYffusion | 2.9   | 1.52  | 0.88 | 3.53   | 1.86  | 0.95 |
| | ARCI-24/1h | 2.66  | 1.35  | 0.91 | 3.37   | 1.73  | 0.96 |
| t2m   | DYffusion | 2.41  | 1.18  | 1.09 | 2.9    | 1.4   | 1.09 |
| | ARCI-24/1h    | 2.07  | 0.95  | 0.91 | 2.57   | 1.18  | 0.94 |
| u10   | DYffusion | 3.65  | 1.89  | 0.87 | 4.1    | 2.14  | 0.92 |
| | ARCI-24/1h | 3.44  | 1.74  | 0.94 | 3.99   | 2.05  | 0.97 |
| v10   | DYffusion | 3.72  | 1.93  | 0.88 | 4.16   | 2.17  | 0.94 |
| | ARCI-24/1h | 3.54  | 1.8   | 0.93 | 4.1    | 2.11  | 0.98 |
| ws10  | DYffusion | 2.51  | 1.36  | 0.91 | 2.68   | 1.47  | 0.95 |
| | ARCI-24/1h | 2.38  | 1.28  | 0.97 | 2.62   | 1.42  | 0.99 |

Although DYffusion achieves much better results than the 1h-GenCast setup AR-1h (Fig. 6 in paper), **it is beaten by ARCI-24/1h in all variables and lead times**. While doing this retraining and evaluation we have learned key things about DYffusion that we highlight below.

**1. DYffusion is hard to train**

DYffusion requires training two networks, one interpolator and one forecaster. To get a probabilistic model, they introduce a layer dropout term in the interpolator that they keep on during inference. This makes the performance sensitive to the dropout rate, which can not be changed without retraining both the interpolator and forecasting networks.

For our model, we took a successful Diffusion framework for images and applied it to weather, with minimal changes to the hyperparameters. We can also tune the performance after training by changing the noise levels in Table 3.

**2. The interpolator has problems with blurring and temporal consistency**

Since all intermediate forecasts are generated using the interpolator, this sets an upper limit on the performance of how well we can expect to perform. The interpolator is trained in the same way as a deterministic models using a MSE loss, which has shown to be prone to blurring.

Further, the interpolator gives no guarantee of temporal continuity of trajectories. The stochastic interpolator (with dropout activated) might predict forecasts at 11h, 12h, that are both possible given the endpoints, but that are not consistent with each other due to sampling different dropout at the different time steps. In our paper we propose a way to correlate the forecasts at each time, overcoming this issue.

[1] Price, Ilan, Alvaro Sanchez-Gonzalez, Ferran Alet, Tom R. Andersson, Andrew El-Kadi, Dominic Masters, Timo Ewalds, et al. “GenCast: Diffusion-Based Ensemble Forecasting for Medium-Range Weather.” arXiv 2024. https://doi.org/10.48550/arXiv.2312.15796.

[2] Cachay, Salva Rühling, Bo Zhao, Hailey James, and Rose Yu. “DYffusion: A Dynamics-Informed Diffusion Model for Spatiotemporal Forecasting,”. Advances in Neural Information Processing Systems, 36, 2023. https://openreview.net/forum?id=WRGldGm5Hz

[3] Cachay, Salva Rühling, Brian Henn, Oliver Watt-Meyer, Christopher S. Bretherton, and Rose Yu. “Probabilistic Emulation of a Global Climate Model with Spherical DYffusion.” Advances in Neural Information Processing Systems, 37, 2024. https://openreview.net/forum?id=Ib2iHIJRTh

---

### Meta-Review · Area_Chair_aDyy · 2024-12-29

**Metareview:**

The paper introduces a new approach for ensemble weather forecasting using diffusion modeling. The method is not autoregressive and can generate ensemble member trajectories from initial conditions without relying on iterative steps, even though it can be combined with autoregressive methods for long-range forecasts. Uncertainty quantification is provided via repeated sampling. The approach is theoretically sound.

**Additional Comments On Reviewer Discussion:**

The authors provided a rebuttal thoroughly answering the reviewers' questions and comments. The authors engaged with the reviewers, most of the time without an answer from the reviewers, and provided new results from baselines such as Dyffusion.

---

### Decision · Program_Chairs · 2025-01-22

Accept (Poster)